# A pharmacoproteomic landscape of organotypic intervention responses in Gram-negative sepsis

Tirthankar Mohanty [1,4], Christofer A. Q. Karlsson [1,4], Yashuan Chao [1], Erik Malmström[1,2], Eleni Bratanis [1], Andrietta Grentzmann [1], Martina Mørch [1], Victor Nizet [3], Lars Malmström [1], Adam Linder [1], Oonagh Shannon[1] ✉ & Johan Malmström [1] ✉

Sepsis is the major cause of mortality across intensive care units globally, yet details of accompanying pathological molecular events remain unclear. This knowledge gap has resulted in ineffective biomarker development and sub-optimal treatment regimens to prevent and manage organ dysfunction/damage. Here, we used pharmacoproteomics to score time-dependent treatment impact in a murine *Escherichia coli* sepsis model after administering beta-lactam antibiotic meropenem (Mem) and/or the immunomodulatory glucocorticoid methylprednisolone (Gcc). Three distinct proteome response patterns were identified, which depended on the underlying proteotype for each organ. Gcc enhanced some positive proteome responses of Mem, including superior reduction of the inflammatory response in kidneys and partial restoration of sepsis-induced metabolic dysfunction. Mem introduced sepsis-independent perturbations in the mitochondrial proteome that Gcc counteracted. We provide a strategy for the quantitative and organotypic assessment of treatment effects of candidate therapies in relationship to dosing, timing, and potential synergistic intervention combinations during sepsis.

Sepsis, a leading cause of mortality in ICUs[1,2], contributes up to ~19% of deaths worldwide[3]. Recent Sepsis-3 guidelines define the syndrome as life-threatening organ dysfunction caused by dysregulated host response to infection[4]. The variability in sepsis complicates delineation of patient subgroups/subphenotypes with similar clinical and molecular features[5–8], which makes it challenging to introduce precision medicine strategies. The systemic nature of sepsis introduces a highly complex pattern of overlapping molecular disease trajectories that can cause organ damage through disparate immunological and metabolic processes[9,10]. In this regard, metabolic deregulation due to aberrant mitochondrial pathways is an important process driving septic organ dysfunction[11–13]. Loss of mitochondrial homeostasis and related

bioenergetic disturbances may explain cessation of cellular functions without excessive cell death[14,15]. Exact mechanisms of this dysfunction have not been elucidated, hindering development of novel mitochondria resuscitating drugs as adjuvant therapies.

Sepsis-induced organ alterations are interconnected and systems-wide[16]. However, knowledge of variations in organ responses due to differences in organ structure and function is limited. Proteomics can provide valuable insights into time-resolved systems-wide and organ-/cell-specific septic proteotypes, i.e., proteome assessment at a particular time across organs[17–24]. Defining organotypic proteotypes in murine sepsis models, where each organ displays uniquely enriched protein signatures, can improve not only understanding of organ-specific

[1]Division of Infection Medicine, Department of Clinical Sciences, Lund, Lund University, SE-22184 Lund, Sweden. [2]Emergency Medicine, Department of Clinical Sciences Lund, Lund University, Skåne University Hospital, Lund, Sweden. [3]Department of Pediatrics, Division of Host-Microbe Systems and Therapeutics, University of California, San Diego School of Medicine, La Jolla, CA, USA. [4]These authors contributed equally: Tirthankar Mohanty, Christofer A. Q. Karlsson. ✉e-mail: oonagh.shannon@med.lu.se; johan.malmstrom@med.lu.se

combinations of damage, inflammation, and immunological and metabolic disease trajectories, but also assessment of treatment impact.

Over the last four decades, >120 sepsis treatment trials have been conducted unsuccessfully despite promising results in preclinical models. Understanding molecular treatment responses introduced by tested interventions in models is limited by readouts like survival and inflammatory markers that offer little insight into organ dysfunction. Strategies that objectively quantify intervention efficacy at a molecular level and define on-/off-target intervention-specific effects are needed. Further, differences in organ-/cell-specific septic proteotypes suggest that treatments could exert non-uniform effects across organs. Currently, it remains unclear to what degree interventions can revert dysregulated proteins introduced by sepsis, how such reversions would vary across organs, and if interventions may introduce iatrogenic changes unrelated to sepsis[5,6]. Lack of adequate bioinformatic tools to objectively score the multi-faceted intervention benefits further complicates evaluation of therapies and definition of optimal treatment windows.

Here, we provide a strategy to define intervention-sensitive proteotypes objectively and quantitatively in mice using a combination of conventional markers of sepsis, proteomics, and systems biology. Using a defined inoculum of *Escherichia coli* in a murine intraperitoneal challenge model and time-resolved deep proteome maps of organ and blood compartments, we generate a previously unknown perspective of in vivo molecular events preceding and associated with organ damage in sepsis. We then develop an unbiased strategy to—at a proteome-wide level—score therapeutic effects of antibiotics and corticosteroids. Scoring interventions revealed time-dependent, intervention-specific, and synergistic effects in an organ- and blood compartment-specific manner during disease. We anticipate this approach will allow researchers to better characterize treatment effects in preclinical sepsis models, define treatment-indicative biomarkers based on species-conserved regulatory networks that can be tracked in human studies, and improve translation of novel sepsis therapies.

## Results

### Multiorgan proteome landscape of murine sepsis and effects of molecular therapeutic interventions

To define the multiorgan landscape of sepsis and uncover molecular therapeutic intervention responses, we applied a combinatorial strategy using pathophysiological read-outs such as cytokines, immune cells, organ damage, and bacterial burden, coupled to a deep sepsis proteome tissue atlas, thus describing the temporal proteome restructuring in plasma and organs at high risk of developing damage[4] (Fig. 1a). C57BL/6 J mice were inoculated intraperitoneally (IP) with $10^4$ colony forming units (CFU) of E. coli strain O18:K1 and the time-resolved systemic molecular sepsis response 0-, 6-, 12- and, 18 h post-infection (h.p.i.; $n = 23$) determined to identify possible therapeutic treatment windows (Fig. 1a–c).We repeated IP *E. coli* inoculation with combinatorial administration of 10 mg/kg meropenem or 30 mg/kg methylprednisolone at 2- or 8 h post-infection (h.p.i.) to examine synergistic, antagonistic, and/or orthogonal molecular intervention effects. Meropenem is a broad-spectrum bactericidal β-lactam antibiotic used in empiric sepsis therapy but with limited knowledge of effects on host proteome responses. The synthetic glucocorticoid methylprednisolone exerts anti-inflammatory and/or immunosuppressive effects, and is frequently studied as a sepsis intervention without consistent benefits[25].

Heart, kidney, liver, lung, spleen, total leukocyte pool, and blood plasma were harvested to construct a murine sepsis spectral library from fractionated organs and cell lysates. This library contained 4,614,190 peptide spectrum matches covering 102,829 peptides matching to 11,510 proteins (Supplementary Fig. 1). 557 DIA-MS proteome maps were obtained from organs, cells, and plasma, and the corresponding spectral library was used to extract peptide and protein quantities, identifying 9620 unique protein groups; all data, including spectral library, 1500 proteome maps and result files are available from ProteomeXchange[26] (PXD036832, PXD036847) (Fig. 1a). To the best of our knowledge, the resource presented here provides the largest compendium of label-free DIA-SWATH maps for an animal model of infection.

### Multiorgan time-dependent monitoring of murine E. coli sepsis

Over time, animals progressively developed severe sepsis characterized by time-dependent reductions in weight (Supplementary Fig. 4), total white blood cell and platelet counts, coupled to a reciprocal increase in neutrophil activation (Fig. 1e). Spleen and plasma, and the heart at later time points displayed elevated inflammatory cytokines (Fig. 1f). Spleen displayed high CFU during early stages (6 h.p.i) (Fig. 1g). Although the proteome data segregated individual organs by sepsis response, their intrinsic tissue proteomes remained distinct and enriched for typical organotypic protein machineries, like cardiac myofibrils in heart, high-density lipoproteins in plasma, and antigen processing in spleen (Fig. 1h).

### Profiling septic organ damage and tissue protein leakage in plasma

Multiorgan failure was assessed using a combinatorial strategy of histology, proteomics, plasma-based organ damage markers (ODMs), acute-phase and immune cell-derived proteins. Sepsis-induced histological changes were organotypic, such as splenic white pulp enlargement 6-h.p.i., reduction of hepatocyte nuclei, mesangial proliferation, and clot formation in all organs (Fig. 2a,b). At later stages, established plasma-based ODMs including lactate dehydrogenase (LDH), blood urea nitrogen (BUN), alanine aminotransferase (ALT), and cardiac troponin T were elevated significantly (Supplementary Fig. 4). Manifestations of organ damage and inflammation were supported by proteome data and revealed time-dependent elevation of protein groups related to fibrin clot formation, neutrophil and platelet degranulation, and interferon-beta (IFN-β) response in plasma and organs, all of which in excess are linked to organ dysfunction[27] (Fig. 2c). Increases in ODMs and inflammatory proteins were accompanied by distinct changes in plasma proteome homeostasis, with a marked rise of organ/cell resident proteins entering circulation (Fig. 2d). In total, 275 tissue-abundant and tissue-specific proteins increased in plasma in a time-dependent fashion (Fig. 2e), including those normally involved in specialized tissue functions such as cell adhesion, glycolysis, gluconeogenesis, glutathione metabolism, and aldehyde metabolic process (Fig. 2f). Several functional groups of leaked proteins were overrepresented in specific organs, such as α-amino acid metabolic process in liver, aldehyde metabolic process in lungs, and glutathione metabolism in heart and liver (Fig. 2g). In contrast to conventional ODMs, >37% ($n = 103$) of the leaked proteins had increased organ abundance during sepsis, paralleling their release into plasma during organ damage (Fig. 2f). Organ damage-associated tissue leakage occurs later (12- and 18 h.p.i.) and is rather stochastic, likely due to protein localization and abundance, ease of release into plasma, and magnitude of damage. Liver is the major contributor to this signal likely due to its size, vascularization, and significant baseline contribution to the plasma proteome. Our results perhaps highlight an underappreciated magnitude of leakage of functionally connected tissue proteins into plasma where they are normally absent and can serve as a future resource to define novel plasma-based ODMs and delineate distinct damage responses.

### Temporal profile of organotypic proteome responses in E. coli sepsis

The observed dysfunction across individual organs is likely to disrupt intricate crosstalk essential for maintaining homeostasis[28,29]. Although

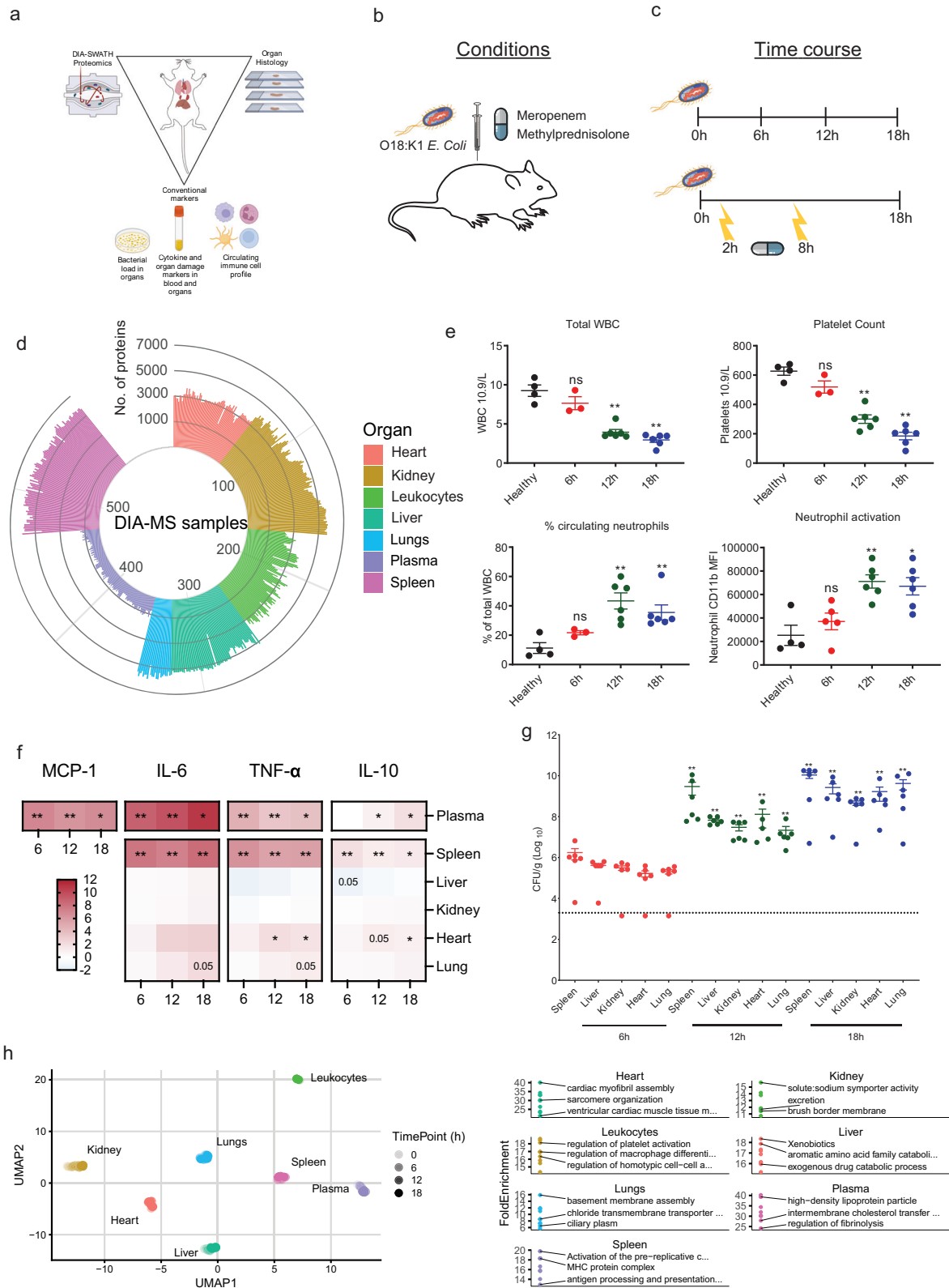

individual organ proteomes segregate over time in an organ-dependent manner during sepsis (Fig. 3a), we lack a basic comparative understanding of organ dysfunction between these organs from infection to severe disease[29], including differences in the onset and rate of decline. We, therefore, performed pairwise differential protein abundance testing to determine when differentially abundant proteins (DAPs) arose and if they persisted with increasing severity. Over the

sepsis course, ramping up of responses (Fig. 3b) and plateauing in the regulation of many of the DAPs at 12 h.p.i. (Supplementary Fig. 5) was seen. DAPs observed in plasma and organs may be due to de novo synthesis, deposition, or reduced clearance. Based on pattern and magnitude, we observed organs displaying bidirectional (liver, leukocytes, and spleen) or unidirectional (lung, heart, kidney, and plasma) regulation of DAPs (Fig. 3b and Supplementary Fig. 5).

**Fig. 1 | Proteomic landscape of murine sepsis. a** Model of sepsis and interventions. C57BL/6J were infected with *Escherichia coli* O18:K1 strain. (Created with bioRender, bioRender.com). **b** Time course of infection and intervention. C57BL/6J were infected with $10^4$ *E. coli* and (**c**) organs were harvested at 0-, 6-, 12- and 18 h.p.i. ($n = 23$ total; 6 for all time points, except time point 0 where $n = 5$) and analyzed with DIA-SWATH maps, organ histology and conventional markers like bacterial load in organs, cytokine profile, organ damage markers and circulating immune cell profile. For the intervention study, infected animals were treated with 30 mg/kg methylprednisolone and 10 mg/kg meropenem ($n = 4$/treatment). **d** Total split of the samples and detected proteins using DIA-SWATH MS. **e** Leukocyte pool during sepsis time course. Total white blood cell count ($10^9$/L), blood platelet count ($10^9$/L), percentage of circulating neutrophils (% of total WBC), and median fluorescence intensity (MFI) of activation marker CD11b on neutrophils as determined by flow cytometry. Data are presented as mean and error bars indicate SEM of mean.

All groups were compared to 0 h by two-tailed Mann–Whitney, *** $p < 0.0005$, ** $p < 0.005$, * $p < 0.05$ and ns = non-significant. **f** Cytokine levels in plasma and organ homogenates depicted as $\log_2$ fold change. All groups were compared to 0 h by two-tailed Mann–Whitney, All groups were compared to 0 h by two-tailed Mann–Whitney, *** $p < 0.0005$, ** $p < 0.005$, * $p < 0.05$ and ns=non-significant. **g** Viable bacterial load in organs. Viability was assessed in spleen, kidney, lung, liver, heart, and spleen. Colony forming units (CFU) expressed as CFU/g of tissue (See Fig. S2 for all combined physiological data). Data are presented as mean and error bars indicate SEM of mean. All groups were compared to 6 h by two-tailed Mann–Whitney, *** $p < 0.0005$, ** $p < 0.005$, * $p < 0.05$ and ns = non-significant. **h** UMAP projection of organs over time. UMAP projection depicting organs segregation over disease progression. The table indicates the functional hallmark groups associated with each organ.

The proteome response can be broadly divided into three categories: DAPs that are uniquely regulated in one organ or plasma, regulated DAPs shared across organs, and uniquely plasma and tissue-regulated DAPs. Clustering DAPs revealed that all organs including plasma were exposed to a synchronized time-dependent increase in fibrin clot formation and neutrophil degranulation (Figs. 2c, 3c,d and Supplementary Fig. 6). Response to IFN-β was observed in organ tissues but not in plasma. Other systemic proteome changes confined to organ tissues included functions like altered Toll-like receptor-4 pattern recognition receptor signalling pathway, insulin pathway regulation, and apoptosis (Supplementary Fig. 6). Further, organs displaying unidirectional regulation of DAPs had minor downregulation in terms of organotypic responses, as the above-mentioned changes represented a notable proportion of the observed response (Supplementary Fig. 7). In contrast, organs with bidirectional regulation were characterized by substantially higher numbers of sepsis-reduced DAPs. Distinct changes in organotypic biological processes include steroid and fatty acid metabolism in liver, altered glucose metabolism and lipoprotein particle composition changes in plasma, and reduced ribosome levels in heart (Fig. 3e, Supplementary Fig. 6 and Supplementary Fig. 8). In addition, we observed time-dependent changes in several hallmark proteins studied in human sepsis such as a reduction of glucocorticoid receptor (GR)-associated pathways in liver including IFN-1 signalling[30], fatty acid metabolism, and gluconeogenesis[31] (Fig. 3g,h). The low-density lipoprotein receptor (LDLR) was also reduced in liver accompanied by increased levels of proprotein convertase subtilisin/kexin type 9 (PCSK9) in plasma, culminating in altered balances between the high-density lipoprotein-associated apolipoprotein a1 (APOA1) and low-density lipoprotein-associated apolipoprotein b (APOB) (Fig. 3i). Concurrently, over the time course, a drop in IGF-1 and insulin-like growth factor binding proteins (IGFBPs) was noted, increasing bioavailability of IGF-1 in plasma (Fig. 3j), with only minor changes in the mitochondrial oxidative phosphorylation system (OXPHOS) (Fig. 3k). We also observed disturbances in protein levels of several proteostasis components in organs and plasma (Supplementary Fig. 9). Alteration of DAPs in terms of tissue-specific and common systems-wide denominators was also examined. DAPs were classified into 4 categories: expressed-in-all, mixed, group-enriched, and tissue-enriched, as described previously[32,33]. Although substantial changes to the proteome were observed, much remain unchanged (Supplementary Fig. 10). Upregulated proteins mostly belonged to expressed-in-all and mixed categories and were involved in inflammatory responses. In contrast, the downregulated DAPs belonged to the group-/tissue-enriched category. These could be linked to rewiring of metabolism and cell cycle in liver and spleen, respectively.

In conclusion, sepsis instigates multiple organ failure and a massive and organotypic proteome reorganization of all analyzed organs in this model. The temporal profiles demonstrate that many responses increase linearly during the early phase of sepsis and plateau during later stages, raising the possibility that early stages represent treatment-sensitive proteostates.

## Definition of proteome intervention responses during sepsis

To test treatment responses, we administered glucocorticoid (Gcc) or meropenem (Mem) alone or in combination at 2- or 8 h.p.i. (Fig. 1a and 4a). At 2 h.p.i., the infection was systemic but with low organ CFU ($10^4$–$10^5$ CFU/g of tissue) and weight loss (Supplementary Fig. 11). By 8 h.p.i., the animals were progressively sicker with a 10- to 100-fold increase in organ CFU and an additional 8% weight loss (Supplementary Fig. 11). The 8 h.p.i time point falls within the time-range before the proteome response starts to plateau (Fig. 3). Mem intervention eliminated *E. coli* CFU in heart, lung, and kidney, with low residual CFU in spleen and liver independent of intervention time (Fig. 4b and Supplementary Fig. 12). Conversely, early Gcc increased *E. coli* CFUs in all organs compared to untreated animals. Interestingly, early Gcc blunted bacterial killing by Mem, possibly by decreasing levels of NADPH oxidase component cytochrome B-245 Beta Chain (CYBB) required for effective oxidative burst in leukocytes[34] (Supplementary Fig. 11). In contrast, late combination intervention resulted in a markedly reduced organ CFU, proinflammatory cytokines in plasma and spleen, weight loss, and elevated WBC levels, together indicating that Gcc therapy accentuates some of the positive effects exerted by Mem (Fig. 4b and Supplementary Fig. 12).

Proteome response profiles supported the above findings. Extensive molecular disturbances instigated by sepsis in our model were complex and acute, and despite systemic effects of interventions, both glucocorticoids and antibiotics could only partially revert the proteome to resemble the uninfected state. Plotting proteome intervention responses both as fold-change sepsis vs. untreated ("Sepsis Response") and fold-change intervention vs. sepsis ("Sepsis Response Reverted") segregated the response into three categories (Fig. 4c): sepsis response—proteins changed by sepsis but unaffected by the interventions (green; no intervention response); sepsis reverted—proteins changed by sepsis but significantly reverted by interventions toward the uninfected baseline (blue; complete reverted); and sepsis-independent treatment—proteins that did not change due to sepsis but only by the treatments (red; sepsis independent) (Fig. 4c,d). On this organ-proteome level, the non-reverted sepsis signal was surprisingly strong, indicating that tested interventions can only partly counteract the sepsis-induced pathways (Supplementary Fig. 13).

Of the tested interventions, Gcc8hMem8h (glucocorticoid and meropenem at 8 h) significantly reverted the most proteins, and to a large degree dampened the sepsis proteome response of non-significantly reverted proteins (Fig. 4d,e). To quantify the intervention impact, we calculated the slope and $R^2$ value (Supplementary Fig. 14) and the percentage of the reverted sepsis response for proteins in the two protein categories for all organ-intervention combinations (Fig. 4f). In this calculation, a slope and $R^2$ of 1.0 equals a complete reversion of the sepsis response by interventions. These calculations

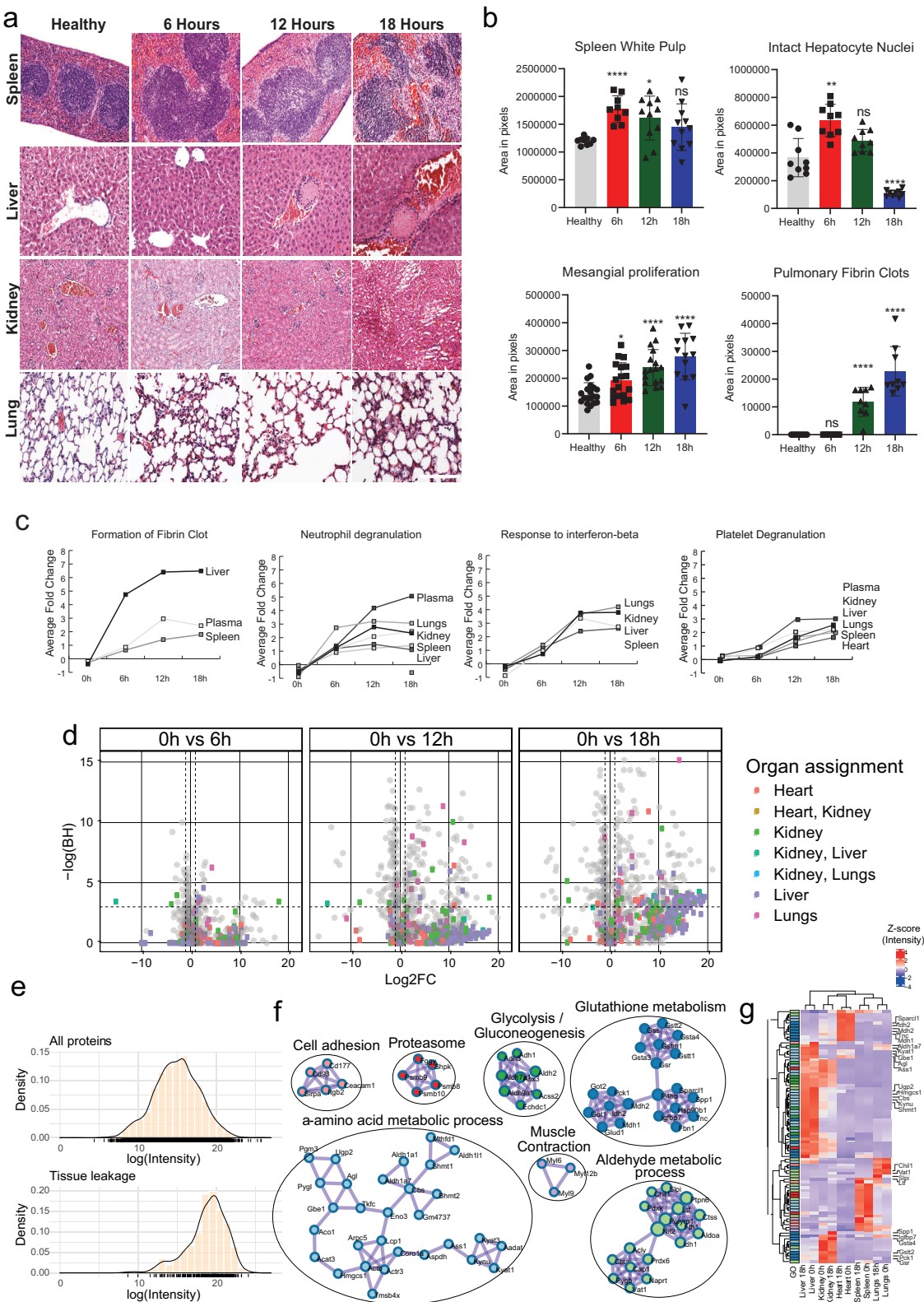

show that late combination of Gcc8hMem8h resulted in the highest average $R^2$ (0.54), the highest slope (0.44), and the largest fraction of reverted proteins from the sepsis response (31%), with the strongest beneficial treatment effect on heart and leukocytes. Gcc2h alone represented the opposite extreme, with minimal reversion of the sepsis response (12%, $R^2 = 0.29$, slope = 0.17) possibly due to high bacterial CFU. This effect was most noticeable in kidney and spleen, where early Gcc2h resulted reverted <10% of the sepsis-induced

proteins. One strong reversion effect of Gcc2h, however, was a significant reduction in the response to IFN-β (Supplementary Fig. 13). To benchmark the magnitude of the quantifiable intervention response, we repeated the experiment including Mem2h as a positive control. Mem2h had a substantially higher average $R^2$ value (0.86) and slope (0.77) compared to the other treatments and reverted on average ~80% of the sepsis-significant DAPs. This highlights the importance of early source control, and that reversal is not absolute since meropenem

**Fig. 2 | Organ damage during sepsis. a** Histology of organ damage. Representative hematoxylin and eosin (H&E) staining of spleen, liver, kidney, and lung. The images depict focal necrosis in the white pulp of spleen (black arrowhead), appearance of clots in the liver (asterisk), tubular necrosis in the proximal tubules in kidney (black arrow), and appearance of clots in the alveoli in lungs (white arrow). **b** Semi-automatic quantification of H&E staining. Images were quantified with FIJI. Bars represent mean, dots represent individual values from 3 animals per time point ($n = 3$) and error bars represent SEM of the mean. All conditions were compared with healthy (0 h) using Kruskal−Wallis with *** $p < 0.0005$, ** $p < 0.005$, * $p < 0.05$ and ns = non-significant (see Fig. S2 for a combined panel of all physiological data). Scale bar = 100 μm. **c** Average increase of inflammatory components like formation of fibrin clot, neutrophil degranulation, response to interferon-beta, and platelet degranulation across organs and plasma. **d** Appearance of tissue leakage proteins in plasma. Volcano plots depict the contributions of various organs to the tissue leakage protein pool. Cut-off for $\log_2$ fold change = +/1.5, −logp = ±4. Colors indicate predicted organ assignments. The proteins were assigned to an organ if they were at least >20-fold more abundant in one organ with a corrected $p$ value > 0.05. **e** Density plot of log organ protein intensities of all proteins and for the tissue-specific proteins identified in plasma. **f** Functional enrichment of leaked proteins. **g** Heatmap showing normalized intensity of the tissue leakage proteins in the organs under uninfected state and after 18 h.p.i. The colors in the first column indicate the functional groups shown in (**f**).

does not treat the dysregulated response directly but rather eliminates the source (Fig. S4d,e). Also, in this experiment, Gcc8hMem8h resulted in a higher $R^2$ (0.65) and percentage of reverted proteins (31%) on average compared to other treatments (Supplementary Fig. 15).

These results show that changes in the proteome landscape can be used to quantify the intervention-impact at an organ level. Intervention effects ranged from 7% reverted proteins with an $R^2$ value of 0.11 to a maximum effect of 85% reverted protein and a $R^2$ value of 0.92. In addition, our analysis revealed a third category of proteins that were unaltered during sepsis but significantly changed because of the interventions during septic conditions: sepsis-independent effects (Fig. 4c,d). This is likely due to side effects that are introduced by Mem and Gcc in sepsis. Proteins in this category are easily overlooked yet likely represent an important parameter in assessing interventions, especially as the number of proteins in this category was surprisingly high (>600 proteins), identified mostly in heart and leukocytes Fig. 4d).

## Scoring of intervention-altered pathways

To further explore functional effects among the reverted proteins, we applied the same proteome scoring shown in Fig. 4 to enriched functional protein categories. This analysis revealed that late interventions had strong organotypic effects on altered inflammatory (Fig. 5a) and metabolic responses (Fig. 5b). Reversion of inflammatory responses was most pronounced in kidneys, with responses such as regulation of cytokine production and immune effector processes, cell killing, and carbohydrate metabolic processes[18] strongly reverted back toward the uninfected state by late treatments (Fig. 5a and Supplementary Fig. 16). Combinatorial effects of Gcc8hMem8h potentiated effects exerted by mono-interventions in kidney (Fig. 5c), with Enrichment analysis against the TRRUST database in kidneys showing that Gcc8hMem8h specifically impacted the pro-transcriptional regulators of inflammation like Nfkb1 ($p = 10^{-3}$), Jun, ($p = 10^{-2.4}$), and Sp1 ($p = 10^{-2.2}$)[35] (Supplementary Fig. 12). These inflammation-related functional categories were reverted in heart to a lesser extent, with no noticeable intervention effect in spleen or liver.

We measured cytokines such as IL-6, TNF-α and MCP-1 in plasma using immunoassays and found their levels reduced by all interventions except early Gcc (Supplementary Fig. 12). A similar trend was seen for IL-6 in spleen, which produces IL-6 during endotoxemia[36]. Organ proteomes also revealed that early Gcc did not alter systemic inflammatory responses in any organ including kidney. GccMem8h elevated circulating leukocytes while reducing neutrophil activation, indicating a reversal of disease[37]; conversely, early Gcc treatments were associated with leukopenia. Early Gcc interventions increased concentrations of innate immune response proteins induced in sepsis, such as high mobility group proteins (HMGB-1, −2)[38] and the interferon-type 1 pathway (Ifih1, ifi204)[39]. These pathways regulate IFN-β responses[39], participate in immune recognition of damage-associated nucleic acids as well as heat shock proteins (HSP90AA1, HSP90AB1)[40], and are involved in activation of interferon regulatory factors −1/−3 (IRF1, IRF3) and STAT1[39,41] (Fig. 5d). Early Gcc also reverted proteins like CEBPB−a transcription factor regulating expression of genes involved in immune and inflammatory responses−and CREB1−a phosphorylation-dependent transcription factor that stimulates transcription upon binding to DNA cAMP response element (CRE)[42].

A marked elevated inflammatory response in all organs, an organotypic reduction of DNA repair and transcription in spleen, and fatty acid, steroid, and retinol metabolism in liver were seen (Fig. 3). Several of these processes were distinctly reverted by late interventions, underscoring the importance of timing when administering interventions (Fig. 5b). Most apparent was the partial reversion of the metabolic module in liver including lipid transport, regulation of cholesterol and steroid metabolism, long-chain fatty acid metabolism, and bile acid metabolic processes. The transcriptional regulatory network connected to this effect was HNF1 homeobox A ($p = 10^{-2.6}$), which is expressed highly in the liver and regulates expression of several liver-specific genes[43]. Repression of lipid metabolic pathways in the liver and elevated lipid metabolites are associated with Gram-negative sepsis[44]. Apart from its primary metabolic function, lipid transport also facilitates clearance of toxic lipopolysaccharide (LPS) through high-density lipoprotein (HDL) particles from circulation in the liver[45]. One of the strongest intervention-reverted proteins in this group was LDLR. Reduced PCSK9 function is associated with increased pathogen lipid (LPS) clearance via the LDLR, a decreased inflammatory response, and improved septic shock outcome[46]. In support of these earlier observations, our data indicate repression of lipid transport proteins in liver of septic animals (Fig. 5e). Proteostasis components described earlier were not reverted with treatments, indicating that its resolution may be a long-term process (Supplementary Fig. 3e).

## Organotypic intervention response networks in sepsis

To further investigate proteins that remained unaltered by treatments and to determine if interventions specifically alter host response, we plotted drug response networks. In these networks, significant intervention-protein associations within the three protein categories were connected with edges and organized into a network (Fig. 5f–h). The graphs show >1092 associated proteins remained statistically non-reverted after all interventions ranging from 39 proteins in heart to 331 proteins in spleen (box in Fig. 5f). Functional enrichment shows a substantial degree of inflammation, such as neutrophil degranulation ($p = 10^{-37}$) and complement and coagulation cascades ($p = 10^{-29}$), could not be reverted by interventions. Interestingly, many of these functional categories were unaltered even after intervention with Mem2h, indicating an intervention refractory response that is triggered early in the sepsis response and that is maintained throughout the course of disease (Supplementary Fig. 16).

In general, Gcc2hMem8h reverted sepsis-induced pathways to a minor extent (Fig. 5a,b). A noticeable exception to this was the specific and significant decrease of 92 sepsis-induced tissue leakage proteins in plasma (Fig. 5a and 5g). These tissue leakage proteins are intracellular proteins statistically enriched from liver ($p = 10^{-19}$), that leak out into plasma during sepsis and are involved in biological functions such metabolism of amino acids and derivatives ($p = 10^{-51}$), amino acid

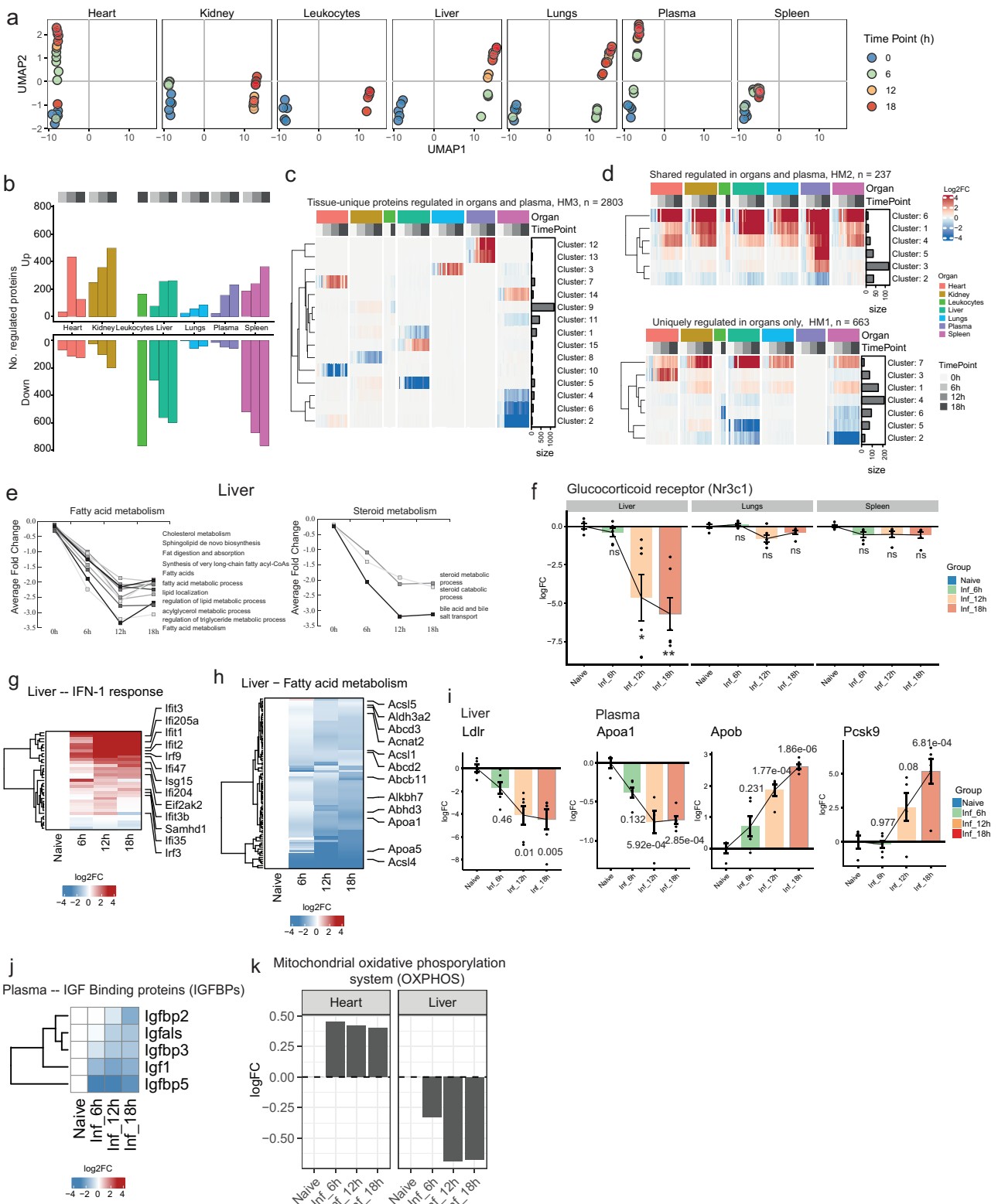

metabolism ($p = 10^{-35}$), and carbon metabolism ($p = 10^{-34}$) (Supplementary Fig. 18). To investigate if Gcc2hMem8h introduced a general reduction in the leakage of tissue proteins into plasma, we plotted intervention trees of the 275 tissue-abundant and tissue-specific that were elevated significantly during sepsis (identified in Fig. 2). These intervention trees show that Gcc2hMem8h cluster with uninfected animals and the gold standard treatment Mem2h (Fig. 5i). In contrast, the increase in plasma inflammatory proteins were unaffected by

Gcc2hMem8h where this intervention clusters with untreated animals (Fig. 5j). As this intervention by itself does not completely clear bacterial CFU, decrease the inflammatory profiles, or reduce organ damage, our results imply that Gcc decreases vascular permeability as indicated by reduced levels of the ODM ALT (Fig. 5k). These findings are relevant because they show that interventions can alter levels of plasma biomarkers and ODMs without affecting inflammation and organ damage.

**Fig. 3 | Temporal profile of organotypic proteome responses in E. coli sepsis.**
**a** UMAP projections of segregated organs over time. **b** The number of differentially abundant proteins (DAP) across organs and the blood compartment over time.
**c** Heatmap showing average normalized intensity of protein clusters that were significantly more abundant on one organ. **d** Upper heatmap in the panel shows protein cluster with significantly increased protein abundances in plasma in at least one organ. The lower heatmap in the panel shows protein clusters with significant increased protein abundances in at least one organ but not in plasma.
**e** Organotypic responses in liver depicting significantly lower levels of proteins associated with functional groups involved in fatty acid and steroid metabolism.
**f** Time-dependent changes in abundance level of the glucocorticoid receptor

(Nr3c1) in liver, lungs, and spleen. Differential abundance against the non-infected controls was calculated using limma (1.11.1) and Benjamini-Hochberg adjusted $p$ values are indicated above as *** $p < 0.0005$, ** $p < 0.005$, * $p < 0.05$ and ns = non-significant. **g** Heatmap showing the increased levels of type-1 interferon (IFN-1) response proteins over time in liver. **h** Heatmap showing reduced levels of fatty acid metabolism proteins over time in liver. **i–k** Time-dependent changes in abundance level regulating apolipoproteins in plasma and liver (**i**) (differential abundance against the non-infected controls was calculated using limma (1.11.1) and Benjamini-Hochberg adjusted $p$ values are indicated above), insulin-like growth factor (IGF-1) signaling in plasma over time (**j**), and the mitochondrial oxidative phosphorylation (OXPHOS) components in heart and liver over time (**k**).

## Characterization of intervention-induced sepsis-independent effects and mitochondrial dysfunction

In addition to reverted/non-reverted proteins groups, we identified a group of proteins that did not change due to sepsis but only by the treatments—sepsis-independent treatment effects. Most sepsis-independent response DAPs were downregulated at the organismal level by all interventions. These DAPs were strongly linked to mitochondria and were enriched for electron transport chain, mitochondrial transport, membrane organization, and translation (Supplementary Fig. 13). Glucocorticoids and antibiotics can influence mitochondrial metabolism and induce stress responses[47–49], but their individual/combined effects on mitochondria have not been described in sepsis. A pronounced presence of sepsis-independent effects DAPs was seen in heart, especially DAPs that were downregulated (>300 proteins) (Fig. 6a). Many functional protein categories in the downregulated sepsis-independent effects response were mitochondria-related (Fig. S6a,b). The two Gcc treatments lowered sepsis-independent effects DAPs (Gcc2h, 79 proteins; Gcc8h, 108 proteins). In contrast, conditions where Mem was added late (Gcc2hMem8h, 315 proteins; Mem8h, 215 proteins) had higher sepsis-independent effects DAPs. Importantly, late combination treatment with Gcc8hMem8h (70 proteins) counteracted sepsis-independent effects DAPs introduced by Mem in heart such as those involved in β-oxidation (Fig. 6b). Their reduction in heart during septic conditions by antibiotics has not been described to the best of our knowledge, whereas the reduction of cell redox homeostasis is known[47]. In line with previous reports demonstrating cardioprotective effects of Gcc[50], our data indicate that Mem in the setting of early Gcc or absent Gcc is associated with cardiac mitochondrial dysfunction.

Because of the pronounced enrichment of mitochondrial proteins in the sepsis-independent effects group, we selected all proteins annotated as mitochondrial proteins from MitoCarta 3.0[51]. This yielded 945 quantified mitochondrial proteins across organs out of 1140 catalogued proteins in MitoCarta, which we plotted based on the three protein categories shown in Fig. 6c (Supplementary Fig. 21). Functional enrichment was performed on up-/down-regulated proteins that belonged to sepsis-independent effects (Figs. S6a,b). These proteins matched to several mitochondrial functions, such as metabolism, mitophagy, signalling and protein import, and calcium signalling, but was most pronounced in OXPHOS and mitochondrial transmembrane transport pathways involved with ATP synthesis (Fig. 6b,c). The DAPs associated with transmembrane transport appear in the small molecule transport group (Fig. 6c). The heart, in particular, is enriched in OXPHOS components to due to its high energy demand[52]. In total, 42 of the known 192 OXPHOS component proteins were altered by the interventions in sepsis (Supplementary Fig. 21). In heart, sepsis itself did not significantly change abundance levels of central OXPHOS core subunits of complex I–V (Fig. 6d). In contrast, all interventions including Mem reduced levels of proteins linked to these complexes by an average of 50% in heart (Fig. 6e). In leukocytes, by contrast, there was an average fold change increase of OXPHOS complexes although not uniform. In liver, effects were treatment-specific and early administration of Gcc increased the relative amount of all complexes

whereas combination treatments lowered the concentration. We also observed a reduction of DAPs associated with cristae that house the OXPHOS components, transmembrane components that are associated with shuttling of ATP between the mitochondrial matrix and the cytosol, calcium homeostasis, and mitophagy. In all cases, GccMem8h had the least sepsis-independent effects (Fig. 6f). Our data demonstrate an organotypic pattern of mitochondrial dysfunction by interventions. Taken together, these results demonstrate that GccMem8h introduced the least number of mitochondrial pathway impairments in the heart.

Damage to mitochondria in the heart during sepsis results in the ultrastructural damage to cristae[14,53]. Our analyses reveal that Gcc was largely beneficial in the heart. However, sepsis-independent effects pertaining to ATP generation and calcium ion transport were still present. In contrast, Mem alone or with early administration of Gcc introduced additional sepsis-independent effects including reduction of β-oxidation, cell redox homeostasis, and metabolism of amino acid and derivatives, all of which are involved in ATP generation[54]. In conclusion, our data suggest that interventions could introduce subtle yet potent changes in the mitochondrial proteome in sepsis, which in turn could influence cardiac function. Thus, sepsis-independent effects DAPs represent a previously unexplored and important category that should be considered to score treatment benefits and probe unexpected drug synergies.

## Discussion

Using a combinatorial strategy of conventional markers, histology, proteomics, and systems biology approach, we demonstrate several unexplored elements of the host response during sepsis and key therapeutic interventions. This approach is more comprehensive than traditional physiological assessments limited to fewer grosser parameters like immunological responses, weight loss, or survival. Previous proteomic strategies rely on a 'triangular' approach—a few targets out of thousands of proteins are chosen to describe pathophysiology and offer the advantage of translation to larger cohorts with high sensitivity. This approach may be limited in defining disease and intervention complexities in sepsis. In contrast, we utilized a 'rectangular' approach, where maximal proteome coverage allowed for discovery of proteostates and scoring of intervention effects, enabling better insight into underlying biology[55]. Recently, sepsis was categorized into endotypes using plasma biomarkers and clinical outcomes to better understand heterogeneity of treatment effects[7]. Based on these findings, the concept of treatable traits in sepsis is the next step that defines actionable therapeutic targets[10]. In addition to sepsis endotypes, our data demonstrate treatment-induced heterogeneity of organ responses, which should be considered as a variable while defining such traits. Further, the strategy outlined here can help in identifying and prioritizing such traits in preclinical models as multiple members of a pathway can be validated.

Although plasma is a source of major interest for organ damage biomarkers in sepsis, events regulating their appearance in plasma is largely undefined. We provide 275 tissue-specific candidate proteins

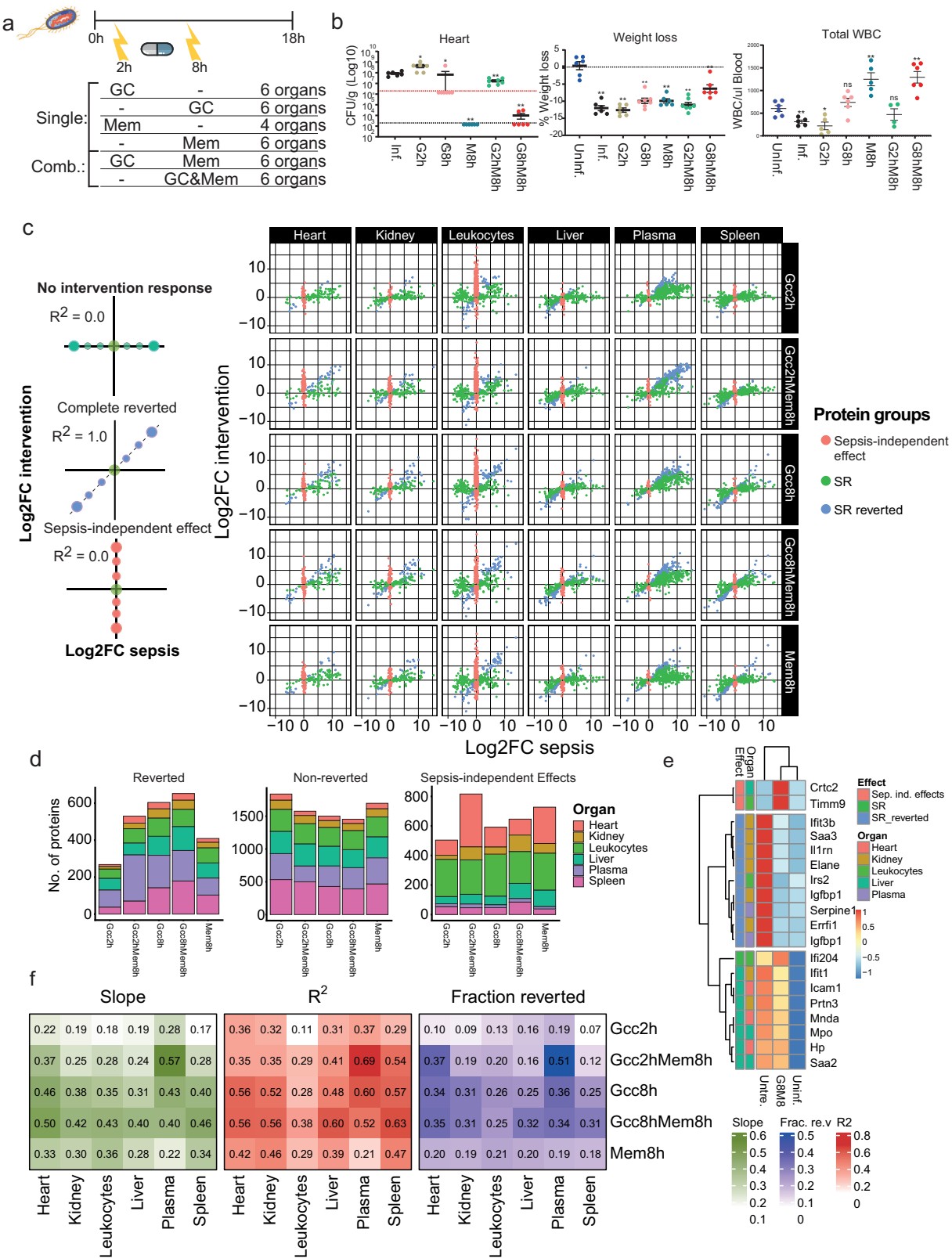

that distinguish between early inflammation and late organ damage. These data may strengthen diagnosis in combination with other inflammatory and ODMs, and perhaps help distinguish between sepsis and other conditions. Many tissue-enriched proteins appear in plasma at 12- and 18 h.p.i. and many are liver-derived. This resembles the sepsis δ phenotype described by Seymour et al.[7], which is characterized by liver failure and elevated IL-6, BUN, and ALT. The liver

has a crucial role in regulating immune and metabolic responses in sepsis[56] and its role in regulating crosstalk in sepsis remains understudied. Apart from being the major contributor to plasma proteome, the liver plays a key in detoxification, modifying insulin sensitivity, secreting soluble signalling molecules and metabolites, and regulating glucocorticoid, amino acid, lipid, and glucose metabolism[57]. Disruption of normal liver function can result in

**Fig. 4 | Scoring intervention effects in organs and the blood compartment.**
**a** Scheme of infection of C57BL/6J and treatment intervention administration.
**b** Physiological markers. Bacterial load in heart (CFU/g of tissue), weight loss (% weight loss prior to infection) and total white blood cell count expressed as $10^9$/L. All animals were harvested at 18 h.p.i. ($n = 6$ per group). Data are presented as mean and error bars indicate SEM of mean. All groups were compared to 6 h by two-tailed Mann–Whitney, *** $p < 0.0005$, ** $p < 0.005$, * $p < 0.05$ and ns = non-significant.
**c** Scheme and scatter plots depicting the three categories of proteome regulation by treatments: reverted, non-reverted, and side effects across organs and the blood

compartment. The colors indicate three protein categories referred to as the reverted (blue), non-reverted (green) and sepsis-independent effects (SIE)(red).
**d** Stacked bar plots showing the number of regulated proteins for interventions in different organs in the reverted, non-reverted and SIE categories. **e** Heatmap of hallmark proteins showcasing the effect of combined glucocorticoids and meropenem administered at 8 h.p.i. (GccMem8h) comparing infected and healthy animals. The first column in the heatmap shows the scoring categories by color.
**f** Heatmaps showing the slope, $R^2$ and fraction of reverted proteins (of total) for the different interventions across different organs.

conditions like lipotoxicity that affects the kidney and heart[44], and accumulation of excess amino acids that can cause brain damage[58]. Our findings have implications for improved characterization of unique host responses over time in organs and the blood compartment and their contribution to organ dysfunction.

Beyond defining organ dysfunction signatures in sepsis, we further used computational strategies to study treatment effects in preventing and reverting this organ damage. We chose the glucocorticoid methylprednisolone and the beta-lactam antibiotic meropenem due to their broad anti-inflammatory and antimicrobial effects, respectively. The combination of interventions administered at 8 h was most beneficial on account of high number of reversions and minimal side effects. Exogenous glucocorticoid therapy downregulates GR in the liver, and increases lipid levels in plasma, both of which are detrimental[59,60]. Drugs enhancing lipid clearance like PCSK9 inhibitors[46] can be envisioned as an adjuvant treatment for glucocorticoids to facilitate lipid clearance in sepsis. Our data also show that pathways describing lipid uptake via LDLR in liver, apolipoprotein B levels in plasma, mitochondrial dysfunction in heart, interferon responses in kidney, and platelet activation in spleen can be envisioned as treatment-indicative markers that are relevant in humans. We believe that some of these can be tracked in human studies to help bridge the translational gap.

Although glucocorticoids do not seem to affect overall mortality in the ICU, they were effective in patients with severe immunosuppression[61]. In our model, mortality was not assessed but interventions reduce the heterogeneity among different individuals on the compartments that perform an immune function (Supplementary Fig. 22), whereas organs that do not perform an immune function and are mainly involved in metabolism are heterogeneous. Thus, the interventions exert a greater effect on responder organs in contrast to bystander organs. Additional experiments are required to validate the use of adjuvant drugs to resuscitate metabolic deregulation and responses need to be benchmarked to study their relevance in mortality. Nonetheless, our approach will enable exploration of the link between timing, dose, drug synergies, and corresponding proteotypes in sepsis.

Cardiac dysfunction is an important component in septic multiorgan failure and corresponding mitochondrial dysfunction is linked to more severe outcomes[14]. There has been a recent resurgence to revisit the introduction of mitochondrial dysfunction by antibiotics[47,53,62]. Our data show that hepatic mitochondrial components including OXPHOS are affected mainly in the liver but not heart in sepsis. In contrast, antibiotics and glucocorticoids introduced cardiac mitochondrial fluctuations in septic animals only. The late combination therapy had minimal downregulation of mitochondrial OXPHOS components and reduced levels of glucose oxidation gatekeeper enzyme pyruvate dehydrogenase kinase 4 (PDK4) in the heart (Supplementary Fig. 23). This indicates that mitochondrial function may be better preserved with late Gcc. Mitochondrial resuscitation could benefit the treatment of septic shock, but due to heterogeneity in organ and individual responses, further studies must be performed to determine which patients would benefit the most. The role of unintended mitochondrial dysfunction in post-sepsis sequelae and identification of surrogate marker that delineates cardiac

mitochondrial status also need to be studied in relevant models and patients. Our strategy can be used to probe drug side effects in other models.

The major limitation of our study was that our study was performed in mice where only parts of the response are similar to humans and, therefore, captured. Further validation studies need to be performed with human material. We chose the human isolate *E. coli* O18:K1 for infection and administered a low dose of $10^4$ bacteria intraperitoneally as described previously[63]. This strategy of inoculation allows the bacteria to gradually grow to a high number, producing observable multiorgan damage and a spectrum of host responses. We saw IFN-β regulatory type-1 interferon responses, fatty acid metabolism, insulin signalling and mitochondrial dysfunction that were benchmarked using accepted pathophysiological parameters including cytokines, circulating immune cell profiles, and ODMs, which have all been described in human sepsis[5,31,64]. Data mining using artificial intelligence of large-scale repositories containing patient and mice proteomes should be analyzed to begin to address the translational gap.

This study presents a resource containing proteomic atlas cataloguing deep time-resolved proteome maps and treatment scoring strategy in sepsis. We define several organotypic signatures of inflammatory and metabolic dysregulation as well as organ crosstalk and damage in sepsis. Further, we characterize treatment effects and provide a scoring strategy that captures reversions, non-reversions, and unintended side effects, thus, improving our understanding of molecular pathogenesis of organ damage in sepsis and molecular treatment effects on the organism. Finally, we present a framework that may be extended to study other diseases, pathogens, treatments, and host response gene knockout functions.

## Methods

### Bacteria and culture conditions
*Escherichia coli* O18:K1[63] from a freezer stock (15% glycerol) was grown in Luria-Bertani broth (LB) overnight (o/n) at 37 °C and 5% $CO_2$ without shaking.

### *E. coli* infection in mice
All animal use and procedures were approved by the local Malmö/Lund Institutional Animal Care and Use Committee, ethical permit number 03681-2019. *E. coli* O18:K1 was grown to an optical density of 0.25 at 620 nm in pre-warmed LB (37 °C, 5% $CO_2$). Bacteria were washed and resuspended in sterile DPBS to $5 \times 10^4$ CFU/ml. Nine-week-old female and male C57BL/6J mice (Janvier, Le Genest-Saint-Isle, France) were infected with 200 µl ($10^4$ CFU) bacteria by intraperitoneal injection. The animals were maintained in a facility with 12 light/12 dark cycle with an ambient temperature of -20 °C. Technicians and staff did not enter the room during the dark cycle unless strictly required to collect cages for health monitoring. The control group was similarly injected with sterile DPBS. Bodyweight and general symptoms of infection were monitored regularly. Mice were sacrificed at 0-, 6-, 12-, and 18 h post-infection (h.p.i., $n = 23$) for the time course, 2- and 8 h.p.i. for the determination of treatment window, and finally all animals were sacrificed at 18 h.p.i. for treatment experiments with antibiotics and corticosteroids. Blood and organs (liver, lung, spleen, heart, and kidney) were collected. Blood was taken by cardiac puncture and

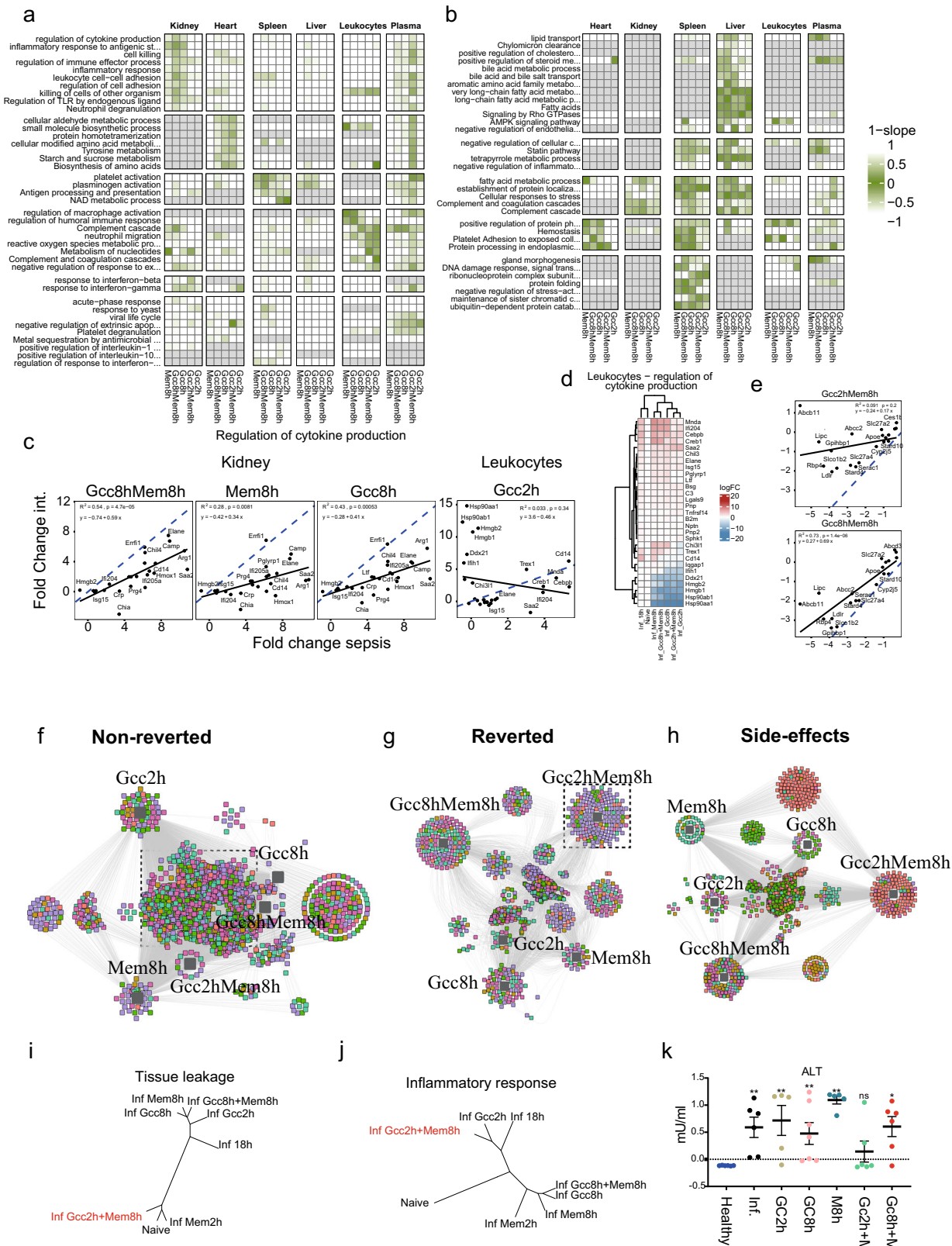

collected in tubes containing sodium citrate (MiniCollect tube, Greiner Bio-One). Methylprednisolone (Solu-Medrol™, Pfizer) and meropenem (Pfizer) were injected intravenously at 30 mg/kg and 10 mg/kg, respectively, at 2- or 8 h.p.i.. Interventions were validated in two independent cohorts (191210_treatment cohort, $n = 44$, $n = 6$/treatment group; 200319_validation cohort, $n = 30$, $n = 6$/treatment group).

**Organ preparations**

Citrated blood collected from infected and control mice was centrifuged (2000 $g$, 10 min) to obtain platelet-free plasma and the cell pellet containing circulating leukocytes was stored for further processing as described later. Plasma was aliquoted and stored at −80 °C. Collected organs were homogenized (MagnaLyzer, Roche) in DPBS

**Fig. 5 | Organotypic intervention response networks in sepsis.** The members of the three protein categories shown in Fig. 4 were subjected to functional enrichment using Metascape[71]. A 1-slope value was calculated for each protein category, where protein categories close to 0 are colored indicating the intervention effect per treatment group shown as a heatmap for functional groups associated with increased protein abundance (**a**) and functional groups with decreased protein abundance (**b**). **c** Example scatter plots of fold-change sepsis vs. fold-change intervention used to calculate the values in (**a**,**b**) for protein members in the functional group 'regulation of cytokine production. Solid line shows the linear regression line, and the dotted line shows a slope of 1, which indicates the theoretical complete reversion all the protein members back to uninfected state. **d** Heatmap showing the normalized protein intensities after intervention for the proteins associated with 'regulation of cytokine production in leukocytes. **e** Scatter plot of fold-change sepsis vs. fold-change intervention for the protein members in the liver for the functional category 'lipid transport' in animals treated with

Gcc2hMem8h or Gcc8hMem8h. The solid line indicates the linear regression line, and the dotted line shows a slope of 1 for comparative reasons. **f–h** The member of the three protein categories related to their respective intervention using the *p* value as edge (cut-off was corrected *p* value of >0.05) and visualized using cytoscape for non-reverted (**f**), and box with dashed lines represents shared proteins for all interventions, (**g**) reverted (dashed box contains the reverted proteins with Gcc2hMem8h), and (**h**) sepsis-independent effect (SIE) by interventions across different tissues. **i** Total tissue-leakage proteins into plasma by different interventions. Colors indicate organ associations. **j** Treatment trees for known inflammatory proteins and the 275 tissue-leakage proteins in plasma for different interventions. **k** Plasma levels of alanine aminotransferase for different interventions expressed as mU/ml. Bars represent mean and error bars represent SEM of the mean. All groups were compared to control by two-tailed Mann–Whitney, *** $p < 0.0005$, ** $p < 0.005$, * $p < 0.05$ and ns = non-significant.

using sterile silica beads (1 mm diameter, Techtum). The homogenates were plated for viable counts and the remaining homogenates were saved for proteomics as described below. In some experiments, intact organs were taken for histology (see below). Bacterial load in organs was determined by serial diluting and plating (heart, lung, liver, kidney, and spleen homogenates) onto blood agar plates. Colony forming units (CFU) were counted following o/n incubation (37 °C, 5% $CO_2$), and are presented as CFU/g of tissue. Remaining organ homogenates were centrifuged (20,000 g, 10 min, 4 °C) and supernatants were immediately transferred and aliquoted into new cryovials. All samples were stored at −80 °C until further analysis. Protein concentration was determined using standard BCA assay (Thermo Scientific) according to manufacturer's instructions.

### Blood collection and flow cytometry analysis
Citrated blood was collected from infected and control mice. For the time-course cohort, full blood counts were determined using an automated hematology analyzer (Abacus vet5). For the treatment cohort, counts for WBCs, neutrophils, and monocytes were determined by volumetric flow cytometry as follows.

Citrated blood was diluted with HEPES buffer containing mouse Fc-block (BD Pharmingen). The following antibodies were used: Alexa Fluor 647 Rat Anti-Mouse CD19 (BD Pharmingen, Catalog No. 557684, Clone 1D3); APC-R700 Rat Anti-Mouse Ly-6G Ly-6C (BD Horizon, Catalog No. 565510, Clone RB6-8C5); FITC Mouse Anti-Mouse CD45.2 (BD Pharmingen, Catalog No. 553772, Clone 104); FITC Rat Anti-Mouse CD41 (BD Pharmingen, Catalog No. 553848, Clone MWReg30); PE Hamster Anti-Mouse CD3e (BD Pharmingen, Catalog No. 553063, Clone 145-2C11); PE Rat Anti-Mouse Ly-6G (BD Pharmingen, Catalog No. 551461, Clone 1A8); and PerCP-Cy5.5 Rat Anti-CD11b (BD Pharmingen, Catalog No. 550993, Clone M1/70). The samples were stained with antibodies (1:200) and incubated for 15 min at room temperature. Samples were lysed using 1-step Fix/Lyse Solution (e-Bioscience) for 30 min at room temperature, washed with PBS (500 g, 5 min), and cell pellets were resuspended in PBS. Two flow cytometry panels were used (see Fig. S1c for representative gating strategies for samples from a healthy and an infected animal 18 h.p.i.). The samples were analyzed using an Accuri C6 Plus flow cytometer and software (BD Biosciences).

### Cytokines and organ damage markers (ODMs)
Cytokines in citrated plasma samples were quantified using the BD Cytometric Bead Array Mouse inflammation kit (552364, BD Biosciences) according to manufacturer's instructions and analyzed using a FACSVerse flow cytometer (BD Biosciences). Plasma levels of alanine aminotransferase (ALT), lactate dehydrogenase (LDH), blood urea nitrogen (BUN), and were measured using ALT Activity Assay Kit (ab105134, Abcam), LDH Assay Kit (ab102526, Abcam) BUN Colorimetric Detection Kit (EIABUN, Invitrogen, Thermo Fisher Scientific), and cardiac troponin I (Abcam, ab285235), respectively. In organ

homogenates, levels of IL-6 (88-7064), IL-10 (88-7105), TNF-α (88-7324), IFN-γ (88-7314), and IFN-α (BMS6027) were measured using mouse ELISA kits (Invitrogen, Thermo Fisher Scientific). All ELISA and colorimetric assays were performed according to manufacturer's instructions in combination with a microplate reader (VICTOR3 Multilabel Plate Reader, Perkin Elmer).

### Pathology scoring and immunohistochemistry of organs
Heart, lung, liver, spleen, and kidney tissue samples from infected and control mice were fixed in Histofix (Histolab Products AB, Askim, Sweden) for 48 h, dehydrated in 70% EtOH for at least 24 h, then embedded into paraffin blocks and sectioned (4 µM microtome, Leica RM2255). Sections were transferred to slides and deparaffinized (60 °C, 30 min) followed by dehydration in Histolab Clear (#14250, Histolab). Sections were stained using Mayer's hematoxylin and eosin (H&E) (Histolab Products AB, Askim, Sweden). Sections were air-dried at room temperature and mounted with Pertex 5 (#00840, Histolab) and a glass coverslip (1.5 mm). Imaging was performed using a light microscope (Nikon Eclipse 80i: ×10 magnification in skin and ×40 magnification in kidney and liver). Representative images from H&E-stained sections of liver (3 images per condition, $n = 3$), spleen (3 pulps per image per condition, $n = 3$), and kidneys were acquired at ×20 and ×40 magnifications. Image analysis of the sections was performed with FIJI version 1.53. The presence of clots in the liver vessels was counted manually. For quantification of kidney dysfunction, mesangial proliferation in kidneys was assessed. A minimum of 36 glomeruli from 17 images ($n = 3$) were selected using the freehand tool and processed similarly as described for spleen above, with Huang chosen as the method for auto threshold. Images were deconvolved using the color deconvolution function, with vectors set to H&E. The image depicting hematoxylin staining was selected and auto threshold set to Otsu was applied. The thresholded image was added to the ROI manager and then objects within the ROI were measured.

### Leukocyte preparation
The blood cell pellet obtained post collection of plasma was transferred to a fresh 15-ml Falcon tube (Sarstedt, Germany) and subjected to erythrocyte lysis using buffer EL (Qiagen, 5 ml) for 10 min on ice, followed by centrifugation at 500 g for 10 min (4 °C, acc/dec = 9). The supernatant was collected carefully without disturbing the pellet and discarded. The process was repeated twice (3 in total), and the leukocyte enriched pellet was lysed in 350 µl RLT buffer (Qiagen) and stored at −20 °C. Cell lysates were processed using the Allprep DNA/RNA kit (Qiagen). The protein-containing flow-through (500 µl) obtained after total RNA isolation was subjected to precipitation using three volumes (1.5 ml) of ice-cold acetone for 1 h in the freezer. Precipitates were then centrifuged for 15 min at 14,000 g at 4 °C. Supernatant was discarded and the samples were air-dried at room temperature for 30 min. The protein precipitates were then dissolved

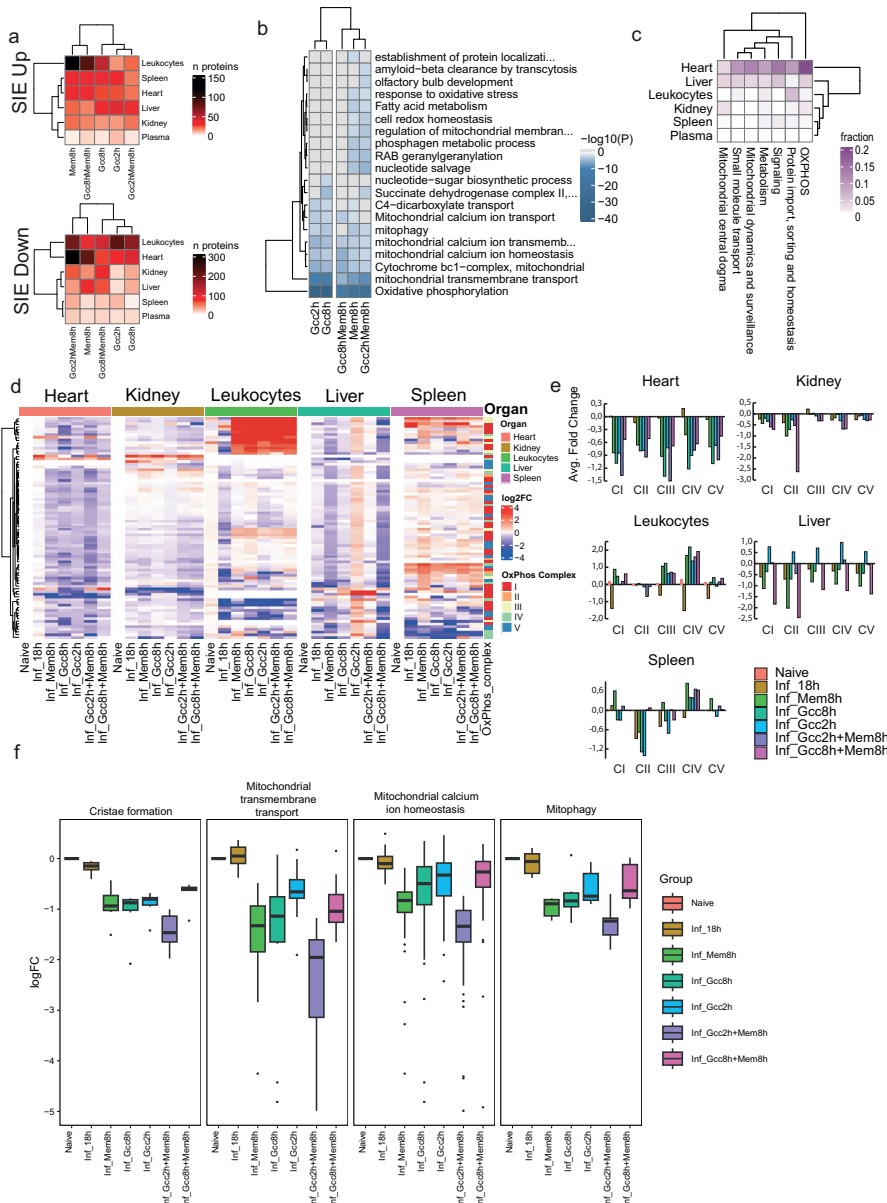

**Fig. 6 | Sepsis-independent effect proteins, enrichment and dysfunction of cardiac mitochondria. a** The number of sepsis-independent effect (SIE) proteins from Fig. 4 per organ and blood compartment separated into reduced or increased abundance levels. **b** Functionally enriched protein categories of the SIE proteins in the heart. Color gradient indicates −log10(P). **c** MitoCarta enriched terms of SIE proteins in organs and the blood compartment. **d** Heatmap showing normalized proteins intensities for the proteins part of oxidative phosphorylation (OXPHOS) proteins in different organs. The color column indicates OxPhos complex I–V. **e** Fold change of the average MS intensities for the OXPHOS complexes in organs.

**f** Box plots of the average MS intensities for the protein members in the mitochondrial terms associated with 'cristae formation', 'membrane transport', 'calcium ion homeostasis', and 'mitophagy'. Colors indicate the type of intervention. Box boundaries represent first and third quartiles, center line indicates median values. The upper whisker extends from the hinge to the largest value no further than 1.5 * IQR from the hinge (where IQR is the inter-quartile range, or distance between the first and third quartiles). The lower whisker extends from the hinge to the smallest value, at most 1.5 * IQR of the hinge. Data beyond the end of the whiskers are called "outlying" points and are plotted individually.

in 50 µl 0.1% RapiGest SF (Waters), 8 M urea (Sigma-Aldrich) and 100 mM ammonium bicarbonate, pH = 8 (Sigma-Aldrich) and stored at −80 °C. 25 µg of protein was taken for in-solution digest as described below. Leukocyte data presented in the time-course experiments were added from an independent cohort.

## MS sample preparation and data acquisition
Plasma, leukocytes, and organ homogenates (liver, kidney, spleen, heart, and lung) were denatured with 8 M urea and reduced with 5 mM Tris(2-carboxyethyl)phosphine hydrochloride, pH 7.0, for 45 min at 37 °C, and alkylated with 25 mM iodoacetamide (Sigma) for 30 min

followed by dilution with 100 mM ammonium bicarbonate to a final urea concentration below 1.5 M. Proteins were digested by incubation with trypsin (1/100, w/w, Sequencing Grade Modified Trypsin, Porcine; Promega) for at least 9 h at 37 °C. Digestion was stopped using 5% trifluoracetic acid (Sigma) to pH 2–3. Peptide clean-up was performed by C18 reversed-phase spin columns according to manufacturer instructions (Silica C18 300 Å Columns; Harvard Apparatus). Solvents were removed using a vacuum concentrator (Genevac, miVac) and samples were resuspended in 50 µl HPLC-water (Fisher Chemical) with 2% acetonitrile and 0.2% formic acid (Sigma). Peptide analyses (corresponding to 1 µg protein) were performed on a Q Exactive HF-X mass

spectrometer (Thermo Fisher Scientific) connected to an EASY-nLC 1200 ultra-HPLC system (Thermo Fisher Scientific). Peptides were trapped on precolumn (PepMap100 C18 3 µm; 75 µm × 2 cm; Thermo Fisher Scientific) and separated on an EASY-Spray column (ES903, column temperature 45 °C; Thermo Fisher Scientific). Equilibrations of columns and sample loading were performed per manufacturer's guidelines. Mobile phases of solvent A (0.1% formic acid), and solvent B (0.1% formic acid, 80% acetonitrile) was used to run a linear gradient from 5 to 38% over 90 min or 120 min at a flow rate of 350 nl/min. The variable window data in-dependent acquisition (DIA) method is described by Bruderer et al.[65] In summary, one MS1 scan with a scan range of 350–1650 m/z, resolution of 120,000, AGC target of 3e6 and maximum IT 60 ms was followed by 26 or 44 DIA MS2 scans with a resolution of 30,000, AGC target 3e6, auto IT and with a stepped normalized collision energy (NCE) of 25.5, 27 and 30. The data dependent acquisition (DDA) method was the manufacturer's default for 'high sample amount'. In summary, one MS1 scan with a scan range of 350–1650 m/z, resolution of 60,000, AGC target of 5e6 and maximum IT 100 ms was followed by top20 MS2 scans with a resolution of 15,000, AGC target 2e5, 100 ms IT and with a normalized collision energy (NCE) of 27. LC-MS performance was quality controlled with yeast protein extract digest (Promega). MS raw data was stored and managed by openBIS (v20.10.0)[66] and converted to centroid indexed mzMLs with ThermoRawFileParser (v1.2.1)[67].

### Spectral library generation

Mouse peptide samples (plasma, spleen, kidney, liver, leukocytes, and heart) from the time-course experiment were pooled per sample type and time points (0 h together with 6 h and 12 h together with 18 h) and then fractionated with High pH Reversed-Phase (Pierce) into a total of 96 samples. The samples were spiked with PROCAL standard peptides[68] and were analyzed with DDA. The resulting MS data was searched with FragPipe (v12.2)[69,70] using default parameters against a protein sequence database containing the mouse reference proteome (EMBL-EBI RELEASE 2019_04), contaminants and PROCAL together with decoys (in total 44824 entries) using, MSFragger (v2.4) and Philosopher (v3.2.3) applying 1% FDR. A spectral library was compiled from the FragPipe output using Spectrast (v5.0), and msproteomicstools (v0.11.0) for retention time (RT) normalization of the PROCAL precursors. The library contained 11,510 proteins, 152,510 precursors, and 152,299 transitions together with corresponding decoys generated with OpenSwathDecoyGenerator v2.4.

### DIA data extraction

A RT normalization library was generated by extracting all albumin precursors from the library generated above and then refined by testing the precursors for data missingness and delta RT scores in a subset of DIA files with OpenSwathWorkflow (v2.4). 59 albumin precursors were finally selected for the final RT normalization library and used for all DIA data extraction downstream. The main searches and scorings were performed by sample type and mouse experiment. DIA data files (n = 568, n biological replicates = 1) were analyzed against the RT and main spectral libraries with OpenSwathWorkflow (v2.4). The OpenSwath data was scored with PyProphet (v2.1.3) utilizing 1% FDR on protein and peptide levels. Finally, the peakgroups were aligned with TRIC from msproteomicstools (v0.11.0).

### Bioinformatics analysis

MS data analysis was performed in R (3.6) with the R package collection Tidyverse (1.3.0) and Bioconductor package manager BiocManager (v3.9). 17,881,324 rows of TRIC output were read into R and decoy precursors were removed. Precursor intensities were summed into protein intensities and the data was normalized with function justvsn() from package vsn (v3.52.0) per sample type and mouse experiment.

Proteins with missing values were imputed with base R stats::runif() function with the assumption of left-censored data missing not at random. For removal of sparsely quantified proteins, the requirement for inclusion at most 1 missing value in 1 sample group. However, in Fig. 2d–f this filter was no applied. The proteomic DIA data was subdivided into three batches corresponding to mouse experiments consisting of time course (180805, Figs. 1–3, 23 animals, 4 groups). Intervention I (191218, Figs. 4–6, 45 animals, 7 groups) and intervention II (200319, Supplementary Fig. 15, 31 animals, 6 groups). Sample groups, quality control, protein counts and data preprocessing are outlined in Supplementary Fig. 2.

Differential abundance testing was performed with R package limma (1.11.1) or with base R function stats::t.test(). Cut-offs were fold change > ±1.5 and adjusted (with method of Benjamini, Hochberg) $p$ value < 0.05. For linear regression scoring of interventions log2FC of sepsis signal (significant proteins of Infected 18 h – Controls) was plotted against log2FC intervention signal (Intervention–Infected 18 h). R2 and slope was calculated witb R stats::lm. Tissue enriched proteins were defined by having at least four-fold higher abundance levels in a particular tissue compared to other tissues (Fig. 1h & Fig. 2d). In fig. S3f protein tissue enrichment was calculated the teEnrichmentCustom function with default parameters from the R package TissueEnrich (1.16.0). Functional and pathway enrichment analysis was performed with Metascape (v33)[71] using the web interface (https://metascape.org/) and the 'Express Analysis of Multiple Gene Lists' workflow. Metascape results were downloaded as 'zip-files' and protein members from terms together with enrichment results were extracted with custom R scripts. In Fig. 1h enriched Reactome pathways and Gene Ontology terms (biological processes, cellular components and molecular functions) were determined with R package clusterProfiler (3.4.4) with default parameters. Network analysis graphs of intervention reverted, non-reverted and sepsis-independent effects proteins generated with R package igraph (1.2.6) and visualized with Cytoscape (3.0.0). UMAP projections were generated with R package uwot (v0.1.4) on data matrices with zero values for undetected proteins in the particular sample. Heatmaps were generated with R package ComplexHeatmap (v2.0.0) using ward.D2 cluster analysis. Upset plot was generated with R package ComplexHeatmap (v2.15.1) using the default parameters ('distinct mode'). The MitoCarta3 mouse database was used for defining mitochondrial proteins and subdivision into functional groups[51].

### Statistical analysis

Statistical analysis was, unless otherwise stated, performed using nonparametric Mann–Whitney tests (Prism v9.1.0 software; GraphPad, Inc). $P$ values < 0.05 were considered statistically significant.

### Reporting summary

Further information on research design is available in the Nature Portfolio Reporting Summary linked to this article.

## Data availability

All proteomics data in the study have been made public. The mass spectrometry proteomics data have been deposited to the ProteomeXchange Consortium via the PRIDE partner repository, PXD036832 (DDA files used in library creation) and PXD036847 (DIA files used for quantification). The repository deposits include MS-sample metadata annotations in SDRF file format (https://github.com/bigbio/proteomics-sample-metadata). Source data used in the main figure panels are also provided. Source data are provided with this paper.

## Code availability

Scripts used for analyses and figure generation of this paper are available at https://doi.org/10.5281/zenodo.7918638.

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

## Acknowledgements

Gisela Hovold, Louise Thelaus, and Jane Fisher are acknowledged for their excellent technical assistance. We would also like to thank Dr. Praveen Papareddy for providing us with the Biodraw PowerPoint templates.

J.M. is a Wallenberg academy fellow (KAW 2017.0271) and is also funded by the Swedish Research Council (Vetenskapsrådet, VR) (2019-01646 and 2018-05795), the Wallenberg foundation (WAF grant number 2017.0271), and Alfred Österlunds Foundation. E.M. is funded by Wenner-Gren Foundation (FT2020-0003), Birgit och Hellmuth Hertz Foundation (F2017/1577) and Svenska läkaresällskapet (SLS-588971). Both O.S. and J.M. are supported by the Swedish Research Council (Vetenskapsrådet, VR) under grant 2018-05795.

Figure cartoons were created with BioRender.com and biodraw templates (Cambridgesoft).

## Author contributions

J.M., O.A.S., E.M., A.L., V.N., L.M., C.A.Q.K., and T.M. conceptualized experiments and designed the study. O.A.S., E.M., E.B., Y.C., C.A.Q.K., T.M., M.M., and A.G. performed experiments and analysis. J.M., L.M., T.M., E.M., and C.A.Q.K performed mass spectrometry analysis. O.A.S., Y.C., L.M., T.M., C.A., Q.K., and J.M. performed data analysis. All authors contributed to the critical editing, presentation of data, writing and formatting of the paper.

## Funding

## Competing interests

The authors declare no competing interests.
