## [Peer review file · Nature Communications]

REVIEWER COMMENTS

Reviewer #1 (Remarks to the Author):

Mohanty et al. A pharmacoproteomic landscape of organotypic intervention responses in Gram-negative sepsis. Intervention-specific organ profiling of Escherichia coli sepsis. How the septic and non-septic proteome is modified by meropenem, glucocorticoid or glucocorticoid + meropenem administration early (2 hrs) vs a mid time point of 8 hrs is also assessed.

This is a strong translational biology report that contains important translational knowledge, to contextualize important molecular events in early sepsis.

<https://www.ebi.ac.uk/pride/archive?keyword=PXD036832>- Data - Sorry, no projects found for search term PXD036832 and the current set of active filters.

Proteomics can provide novel insights into time-resolved organ- or cell-specific septic proteotypes, i.e. an assessment of the state of the proteome at any particular time or proteostates - Proteostasis is the dynamic regulation of a balanced, functional proteome – does your study offer any insight as to how this is orchestrated? Any novel mechanistic insight can be gained from your study?

1. Please reduce introduction by ~400 words (currently it is just under 900 words) – and focus the introduction on proteomics of sepsis – to contextualize your study within the field (human and animal models)

2. “recent reports suggest that metabolic deregulation due to aberrant mitochondrial pathways” – please note this is NOT “recent” – mitochondria have been known to play a role in sepsis since the 1950’s (I am sure if you look before that – you will find evidence that this is not a novel concept) – also please note papers from the 50’s that already show effect of corticosteroids on this response.

PLAGUE TOXIN: ITS EFFECT IN VITRO AND IN VIVO.

RUST JH Jr, CAVANAUGH DC, KADIS S, AJL SJ.

Science. 1963 Oct 18;142(3590):408-9. doi: 10.1126/science.142.3590.408.

Effects of bacterial endotoxin on metabolism. I. Carbohydrate depletion and the protective role of cortisone.

BERRY LJ, SMYTHE DS, YOUNG LG.

J Exp Med. 1959 Sep 1;110(3):389-405. doi: 10.1084/jem.110.3.389.

Shock induced alterations of mitochondrial energy-linked functions.

Mela L, Bacalzo LV Jr, White RR 4th, Miller LD.

Surg Forum. 1970;21:6-8.

Alterations of mitochondrial structure and energy-linked functions in hemorrhagic shock and endotoxemia. Mela LM, Miller LD, Bacalzo LV Jr, Olofsson K, White RR 4th. Adv Exp Med Biol. 1972;33(0):231-42. doi: 10.1007/978-1-4684-3228-2_24.

3. Is there any evidence that mitochondrial 'resuscitation' may improve mortality? It certainly improves organ function (acutely) but interesting reports show that mitochondrial resuscitation might not improve ultimate mortality. Furthermore – interesting hypothesis regarding maladaptive versus adaptive responses suggests that being able to “shut down” non-essential bioenergy metabolism/consumption is associated with improved outcomes from sepsis - the classic example is Phil Dellinger's study of LMNA – iNOS inhibition – in sepsis. The study was stopped early because of increased mortality in the patients who received the iNOS inhibitor – showing that inhibiting NO mediated inhibition of electron transport chain and ATP production results in increased mortality from sepsis. I think your data may provide some very important insight into these mechanisms – the human clinical data does not favour overall improved mortality if organs are prevented from shutting down their bioenergy requirements to conserve energy.

4. What is the mortality in the model? E. coli $10e^4$

5. What is UMAP2? Please specify what organ specific data was used to generate UMAP projection of organs over time.

6. Lung Histology – figure 2A – You may want to choose better representative photomicrographs of the lung. The healthy lung looks like significant alveolar collapse with areas of PMN infiltration and thickening of the alveolar capillary membrane – this “healthy” lung is not a good example of health. (I like the use of fibrin deposition to mark worsening lung injury – traditionally this is associated with increased alveolitis – PMN infiltration – I don't see that in the chosen images – maybe you can pick a more representative photomicrograph that shows classic features of lung injury)

7. This is not clear to me: The pattern and magnitude of DAPs subdivided organs into two groups that we refer to as responder organs (liver, leukocytes, and spleen) or bystander organs (lung, heart, kidney, and plasma) – WHY? Sometimes “more” is just more – how do you know the relative 'weight/significance' of specific DAB in a specific pathway (e.g. a rate limiting enzyme would clearly have a much more profound effect on a path that multiple non-essential changes) or on the trajectory of sepsis?

8. Page 11 - It is also unclear to this reviewer what the authors mean by: This subdivision may be connected to the known activation of the leukocyte population (Fig 1E, CD11b+) and changes in metabolisms during sepsis?

9. Response to IL-1b and TLR4 – confined to organ tissues only – considering IL-1b and TLR4 share the intracellular domain – it is not surprising that they also share response pattern – Maybe you can select

another receptor mediated pathway – that is confined to ‘organ’ specific response to better illustrate your point?

10. “breakpoint between 6-12 h where the increase in response is no longer linear” – What does this time point signify? Is this the time it takes for ‘de-novo’ protein synthesis? Does this time coincide with bacterial clearance?

11. The subdivision of DAP types: Would it not be much clearer to subdivide DAPs into 2: Organ specific versus shared between organs (include plasma as an organ). And for EACH subgroup have 2 categories – shared with plasma or not?

12. “bystander organs were relatively refractory to intrinsic organotypic alterations” – what does this mean? In the context of the number of DAPs for each organ – I am not quite sure I understand what you are trying to convey?

13. Can you provide an idea of the denominator – i.e. each organ must have a “portfolio” of potentially ‘alterable proteins’ – for each organ the number of DAPs you found represents a subsection of the available proteome (Figure 3B). What is that? Maybe rather than showing absolute number of DAPs it would make more sense to present the data as the percent proteome? And even then – would be important to know if the DAPs/available proteome – specific categories of proteins were preferentially altered? To that end – Figure 1H makes a lot of sense to me and is helpful in contextualizing the changes post-infection – but in the absence of a denominator – it is unclear whether fold changes in basement membrane assembly related proteins is because the basement membrane was truly targeted or is this because there is so much basement membrane – or membrane associated proteins in the lung compared to other organs. This is particularly important in the case of the lung for example – where over 40 cell types are present – each expressing somewhere between 40-60% of their genome in relatively small amounts because their numbers may be so small – compare this to something like heart or liver – where the heterogeneity of the tissue is smaller and consequently the potential organ-specific proteome smaller and all cells may be contributing to the same proteome

14. Figure 3D – rather than clustering by organ – would it not be much more clinically relevant to cluster by function? Previous figures have already addressed the issue of change in number of DAPs – I think this figure should underscore if different “functions” are activated over time – for example – up-regulation of ligands, is followed by upregulation of intracellular signaling MAKs, followed by Transcription factors, followed by cytokine/chemokines de novo gene synthesis, followed by upregulation of receptors and changes in cytoskeletal proteins (this is just an example.....)

15. Please reference - hallmark proteins – how were these selected? How were they defined as Hallmark (is this from metascape’s definition?)

16. Figure 3K – not a lot of change in the expression of OXPHOS proteins – this is such a great result! It really points to the fact that some proteins are not regulated at the abundance level. If you pair this data with the increase in iNOS activation – or the expression of other mitochondrial functional inhibitors it is quite clinically relevant. This could also be paired with data showing a shift from aerobic to anaerobic glycolysis, and potentially the shift to expression of fetal proteins in the heart – e.g. shift in expression away from aMHC and increase the expression of bMHC to compensate for electron transport chain “poisoning” by NO (for example).

17. Page 12: An example of how you contextualized changes of clinical relevant is the LDLR/PCSK9/APOA1 and B shown in figure 3I – I think this makes the data immediately relevant to translational biologists.
18. Proteome “plateaus” at 12 hrs – begins to plateau at 8 hrs – what is the relationship between this and bacterial clearance? From your data – in untreated mice there was very little bacterial clearance that happened in organs in the first 18 hrs – did you do CFUs in blood? Can you measure circulating endotoxin and its relationship to the proteome? (do you pick up endotoxin in your proteomics analysis?)
19. Page 15 - “highlighting the importance of early treatment (Figure S4d and Figure S4e)” – I would change this by stating the importance of early source control – since sepsis is the dysregulated response to infection and meropenem does not treat the dysregulated response but presumably eliminates the source.
20. Page 15 - In addition, our analysis revealed a third and more unexpected category – I would eliminate this sentence – given all the side effects of antibiotics and steroids – it is not possible that this comes as a surprise. Again – rather than underscoring the total number of proteins (this may go in brackets) it would be more helpful to understand the proportion this represents of the available proteome.
21. Page 15 – Reversal of inflammatory response was most pronounced in the kidney – does this reflect concentration of antibiotics and corticosteroids eliminated through the kidneys – do they actually see more of the drug?
22. Page 18 - Functional enrichment shows that there is a substantial degree of inflammation, such as neutrophil degranulation ($p = 10^{-37}$) and complement and coagulation cascades ($p = 10^{-29}$), that could not be reverted by any of the tested interventions. – This is nice data. And it makes so much sense – given the mortality from sepsis remains around 20-40% despite our best efforts – the 1092 proteins - what proportion of the sepsis proteome remains unchanged after treatment? Are there categories of proteins that remain preferentially unchanged?
23. Page 18 - interventions can alter the levels of plasma biomarkers and organ damage markers without affecting inflammation and organ damage. – I think here would be very relevant to link back to the emerging concept of “treatable traits” – moving forward sepsis will be treated with a ‘mixture’ of agents that target multiple ‘treatable traits’
24. Page 19 - These DAPs were strongly linked to mitochondria and were enriched for electron transport chain, mitochondrial transport, membrane organization, and translation – this is most rewarding – exactly what would have been expected – nice!
25. However, no clear signs of mitochondrial damage were observed except hydropic mitochondria and perturbed cristae. (Takasu et al., 2013) – Other studies however have documented by EM sepsis induced ultrastructural changes in mitochondria from mice that developed moderate to severe myocardial depression post cecum ligation and puncture with widespread swollen mitochondria with ruptured outer membranes, matrix paling and disrupted cristae associated with vacuolation of the internal compartment DOI: 10.1097/CCM.0b013e31824e1370 – I suspect your H&E is not sensitive enough to identify changes.

26. Discussion – please include in the discussion, a comment on findings re: mitochondria – this is an old topic that is receiving emerging attention: (i) Bactericidal Antibiotics Induce Mitochondrial Dysfunction and Oxidative Damage in Mammalian Cells (10.1126/scitranslmed.3006055); Do antibiotics cause mitochondrial and immune cell dysfunction? A literature review (10.1097/CCM.0b013e31824e1370); Antibiotics Friend and Foe: “From Wonder Drug to Causing Mitochondrial Dysfunction, Disrupting Human Microbiome and Promoting Tumorigenesis” (10.4236/ijcm.2018.93016);
27. In the discussion – return to the point you make in the introduction – how does your study define treatment-indicative biomarkers based on species conserved regulatory networks that can be tracked in human studies and improve the translation of novel sepsis therapies from preclinical models to humans.
28. In the discussion – carefully attention should be paid to contextualizing why Gcc blunted bacterial killing – and what could be potential “markers” that may be used to understand the “proteostate” associated with high-risk of unwanted side effects.
29. In the discussion – special attention should also be paid to the inability of steroids and/or antibiotics to “revert” the septic-phenotype – your data suggests this is not a feature of host heterogeneity (quality of the variability), but rather a feature of complexity (quantity – number of variables). If your data is correct – even correctly assigning patients into specific endotypes will not allow us to completely reverse the septic phenotype because current pharmacotherapy cannot address the entirety of the complexity. Accordingly new treatments are urgently needed - that adequately tackle major gaps – treatable traits – that are not being treated with our current approaches.
30. In the discussion – It would be important to also address the issue of iatrogenicity and how and how your data may be used in the development of harm reduction strategies.

Reviewer #2 (Remarks to the Author):

Combining conventional markers, histology, proteomics, and systems biology, the authors performed pharmacoproteomic analysis on a murine E. Coli sepsis model by scoring time-dependent treatment impact after Mem and Gcc were administrated. They found three organ-specific proteome response patterns, several novel elements of the host response during the course of sepsis and key therapeutic interventions. Overall, this manuscript is well written and comprehensive. Below are some minor suggestions, mainly about the proteomic data, for the authors to consider before publication.

1. Please complete the code for “under accession code XXXXX”.

2. Since this paper is based on the proteomic data, I would suggest providing a bit detailed description about data QC such as missing value rate (e.g., per sample and overall), batch/study design, replicate information if any, etc.
3. "Missing values were imputed with the assumption of left-censored data missing not at random." The authors could explain here a bit more why this assumption holds for their proteomic data. In fact, proteomic data are subject to both missing not at random and missing (completely) at random. And it is known that SWATH-MS data have a lot of missing values.
4. The authors mentioned that batch-effect correction approach was to utilize protein log₂ foldchange in relation to naïve samples per experiment. But there is no information about how batch correction was done and how the experiment/batch was designed.
5. According to Fig.1D, different organs have different numbers of proteins. Does this mean that Fig. 1H was plotted with the overlapping/common proteins? The same question for all other plots where multiple organs are present.
6. Probably an overview heatmap showing the proteomic data of all organs will reveal organ-specific missingness pattern, organ-specific proteins, house-keeping proteins, and proportion of missingness.

Reviewer #3 (Remarks to the Author):

This manuscript is ambitious and will have a wide breath impact on the longitudinal characterization of murine models of sepsis. The text is well-written and despite the highly technical scope of approach and results, it will be easy to understand by non-experts in MS-proteomics.

Concerns:

1.

The authors wrote: "Our findings establish the first-ever quantitative and organotypic assessment of treatment

effects of candidate therapies in relationship to dosing, timing, and potential synergistic intervention combinations during sepsis"

This claim is more pompous than accurate. The effort presented here is unique in its kind given the amount of proteomics data collection, but is not the first longitudinal study of proteome remodeling in septic mice. It is also not the first study to use proteomics to assess organ damage or response drugs in murine sepsis. The authors should tone down their claims and acknowledge the following papers:

PMID: 31568522; PMID: 32037835; PMID: 35349828; PMID: 35913192; PMID: 31604940.

2.

The area of computational analysis of quantitative proteomics is currently well-established and most experts in the field use the same software, which has been stress-tested and shown to deliver accurate results. The authors of this manuscript should explain why they chose to use non-stream computational tools.

3.

The manuscript suffers from a lack of orthogonal validation of at least some of the quantitative calculations. This is important due to the use of uncommon computational tools.

4. There are typos here and there that should be taken care of.

We thank all reviewers for their time and comments. This process has enriched our perspective
of the data and septic organ failure. Based on the comments suggested by the reviewers, we
have now reanalysed our data, provided new supplemental figures, modified existing text, and
added new text. We have added 4 new supplementary figures, S1b, S3e, S3f, and S6e. The new
data and textual additions highlight pre-processing and quality control assessments of the data-
independent SWATH-MS (DIA-SWATH MS), perturbations of proteostasis and energy metabolism,
and alterable proteome in sepsis. We have also changed the format of the whole blood FACs
analysis data in adherence with journal requirements. The platelet data has been removed as the
settings were not optimized for platelet counts. The gating strategy has been added to the
supplement (Fig. S1c) and updated figure panels have been added (Fig. 4, Fig. S4b). We feel that
we addressed all the comments raised by the reviewers and the new additions have improved
the quality of the manuscript substantially. All changes in the manuscript have been highlighted
in red.

Reviewer #1 (Remarks to the Author):

Mohanty et al. A pharmacoproteomic landscape of organotypic intervention responses in Gram-
negative sepsis. Intervention-specific organ profiling of Escherichia coli sepsis. How the septic and
non-septic proteome is modified by meropenem, glucocorticoid or glucocorticoid + meropenem
administration early (2 hrs) vs a mid time point of 8 hrs is also assessed. This is a strong
translational biology report that contains important translational knowledge, to contextualize
important molecular events in early sepsis.

1. <https://www.ebi.ac.uk/pride/archive?keyword=PXD036832>- Data - Sorry, no projects found for
search term PXD036832 and the current set of active filters.

Reply – We apologise for the lack of clarity in the cover letter, the dataset has not been made
public and the above error message is a consequence of this. The reviewer can access the raw
files by clicking on the login link provided above the search bar on the introductory PRIDE
webpage (right-hand corner) and then logging in using the credentials; User -
reviewer_pxd036832@ebi.ac.uk, Pass- WQTCityU.

2. Proteomics can provide novel insights into time-resolved organ- or cell-specific septic
proteotypes, i.e. an assessment of the state of the proteome at any particular time or
proteostates - Proteostasis is the dynamic regulation of a balanced, functional proteome – does
your study offer any insight as to how this is orchestrated? Any novel mechanistic insight can be
gained from your study?

Reply - This is an interesting question that we overlooked in the first version of the manuscript.
We have now reanalysed the data and found indications of imbalanced proteostasis based on
reduced ribosome content, endoplasmic reticulum (ER) stress sensors, increased levels of both
intracellular and extracellular chaperones, and reduced levels of receptors that are involved in
clearance of plasma proteins. We have added these to results section and provided a new
supplemental figure (Figure S3e) to show case these finding. In summary, our experiments
revealed the following:

- • The sensor for unfolded proteins in the ER, Inositol-requiring protein 1 (IRE1a)¹ was
upregulated in all organs over time. In addition, the Interferon-induced, double-stranded
RNA-activated protein kinase (EIF2AK2) that stalls ribosomes by phosphorylating eIF2a²
and activates the integrated stress response mediator cyclic AMP-dependent
transcription factor ATF-4 (ATF4)^{3,4}, was also upregulated in most organs except the
heart and plasma. This suggests that the systems wide initiation of cytoprotective

measures like the unfolded protein response (UPR)⁵ is necessary to counter sepsis-
induced stress in all organs (Fig. comment_2_proteostasis).

- • Our data indicates that the heart contained the highest numbers of markers of
imbalanced proteostasis. We observed a general reduction of most cytosolic ribosomal
components in the heart. Ribosomes are central in the maintenance of a functional
proteome and constitute the first step of proteostasis⁶. Also, different isoforms of HSP90
(HSP90aa1, HSP90b1 and HSP90ab1) that are responsible for protein folding⁷, and
proteasome assembly chaperones (PSMG1-4) that promote the assembly of the 20S
proteasome^{8,9} were elevated. As cardiomyocytes have limited regenerative capacity, the
activation of processes like the unfolded protein response (UPR) can limit organ damage
in the heart and preserve organ function post-sepsis¹⁰.
- • We show in Figure 4 an increase of acute phase proteins in blood plasma like serum
amyloids (SAA) that can aggregate and deposit in tissues as amyloid fibrils¹¹. In order to
protect the secreted proteome, we believe that induced levels of several extracellular
chaperones like haptoglobin (HP), alpha-2-macroglobulin (A2M), clusterin (CLUS), heat
shock protein beta-1 (HSPB1), progranulin (GRN), and serum amyloid P-component
(APCS) in plasma and tissues¹² could represent an active countermeasure for the
formation of amyloid fibrils.
- • Apolipoproteins and desialylated plasma proteins are cleared from the circulation
through the low-density lipoprotein receptor (LDLR) and asialoglycoprotein receptor-1
(ASGR1) respectively^{13,14}. The levels of ASGR1 were downregulated in the liver during
sepsis. We show in Figure 3 that the interplay between the level apolipoproteins, LDLR
and PCSK9 (Proprotein convertase subtilisin/kexin type 9) in the manuscript (Figures 3
and 5). The lack of both receptors on the liver would result in the failure to clear non-
functional plasma proteins leading to a loss of proteostasis in circulation. Thus, this

highlights how proteostasis occurs across the plasma compartment and liver during
sepsis.

Interestingly, in our treatment cohorts we only observe a mild reversal of systemic indicators of
the integrated stress response, even with the best treatment option in our study, GccMem8h
(glucocorticoids and meropenem administered @8h). The lack of effective treatment response
indicates that the stress responses that lead to disturbed proteostasis in sepsis may require a
longer time to resolve even in presence of treatments. Future experiments where long-term
survival is assessed will help shed light on the relationship between treatments, recovery of
proteostasis and overall health.

In line 196 we have added – “We also observed disturbances in protein levels of several
proteostasis components in organs and blood plasma (Figure S3e).”

In line 313 we have added – “Proteostasis components described earlier were not reverted with
treatments, indicating that its resolution may be a long-term process (Figure S3e).”

3. Please reduce introduction by ~400 words (currently it at just under 900 words) – and focus
the introduction on proteomics of sepsis – to contextualize your study within the field (human
and animal models)

Reply – As suggested, we have now reduced the introduction to contextualize the role of
proteomics of sepsis in humans and animal models.

3. “recent reports suggest that metabolic deregulation due to aberrant mitochondrial pathways”
– please note this is NOT “recent” – mitochondria have been known to play a role in sepsis since
the 1950’s (I am sure if you look before that – you will find evidence that this is not a novel
concept) – also please note papers from the 50’s that already show effect of corticosteroids on

this response.

PLAGUE TOXIN: ITS EFFECT IN VITRO AND IN VIVO.

RUST JH Jr, CAVANAUGH DC, KADIS S, AJL SJ.

Science. 1963 Oct 18;142(3590):408-9. doi: 10.1126/science.142.3590.408.

Effects of bacterial endotoxin on metabolism. I. Carbohydrate depletion and the protective role
of cortisone.

BERRY LJ, SMYTHE DS, YOUNG LG.

J Exp Med. 1959 Sep 1;110(3):389-405. doi: 10.1084/jem.110.3.389.

Shock induced alterations of mitochondrial energy-linked functions.

Mela L, Bacalzo LV Jr, White RR 4th, Miller LD.

Surg Forum. 1970;21:6-8.

Alterations of mitochondrial structure and energy-linked functions in hemorrhagic shock and
endotoxemia. Mela LM, Miller LD, Bacalzo LV Jr, Olofsson K, White RR 4th. Adv Exp Med Biol.

1972;33(0):231-42. doi: 10.1007/978-1-4684-3228-2_24.

We have now modified text in the introduction to highlight this in line 56 - "In this regard,
metabolic deregulation due to aberrant mitochondrial pathways is an important process driving
septic organ dysfunction. Loss of mitochondrial homeostasis and related bioenergetic
disturbances may explain cessation of cellular functions without excessive cell death. The exact

mechanisms of this dysfunction have not been elucidated in detail, hindering development of
novel mitochondria resuscitating drugs as adjuvant therapies.”

We have also added relevant references in line 352 that describes that broad treatments like
glucocorticoids can affect mitochondrial function in endotoxemia.

4. Is there any evidence that mitochondrial ‘resuscitation’ may improve mortality? It certainly
improves organ function(acute) but interesting reports show that mitochondrial resuscitation
might not improve ultimate mortality. Furthermore – interesting hypothesis regarding
maladaptive versus adaptive responses suggests that being able to “shut down” non-essential
bioenergy metabolism/consumption is associated with improved outcomes from sepsis - the
classic example is Phil Dellinger’s study of LMNA – iNOS inhibition – in sepsis. The study was
stopped early because of increased mortality in the patients who received the iNOS inhibitor –
showing that inhibiting NO mediated inhibition of electron transport chain and ATP production
results in increase mortality from sepsis. I think your data may provide some very important
insight into these mechanisms – the human clinical data does not favour overall improved
mortality if organs are prevented from shutting down their bioenergy requirements to conserve
energy.

Reply – We thank the reviewer for their comment and have now added text to the discussion and
provided a new supplemental figure (Figure S6e). Previous reports indicate that the energy and
oxygen demand placed by various immune system processes on the host metabolism is
abnormally high during the acute phase of sepsis. Eventually, the disparity between enhanced
energy demand versus limited energy supply results in reduced ATP generation in mitochondria,
downregulation/functional impairment of mitochondrial OXPHOS components and employment
of alternative methods of ATP generation such as glycolysis, beta-oxidation of fatty acids and

pentose phosphate pathway (PPP). This leads to reduced metabolism and energy production, and
deterioration of organ function¹⁵⁻¹⁷. This has also been hypothesized to be a protective strategy
for organs¹⁸. Using the questions above as a starting point we have searched our data for
evidence that captures some of these key energy utilization pathways that directly regulate
mitochondrial function during sepsis as summarized briefly below:

- • In plasma, we show a reduction of insulin-like growth factor-binding proteins (IGFBPs)
that transport insulin-like growth factor 1 (IGF-1) to tissues (Figure 3J). IGF-1 complexed
to IGFBPs regulates glucose levels in sepsis¹⁹ and supplements insulin activity²⁰. A
reduction of these can be linked to the underlying state of insulin resistance. In Figure 3H
and I, we also show a loss of the fatty acid metabolism components and low-density
lipoprotein receptor (LDLR) in the liver. A loss in both will lead to the build-up of
cholesterol-rich LDL particles in the plasma and this again is indicative of insulin
resistance²¹. Both examples indicate the general lack of glucose availability to fuel the
OXPPOS due to insulin resistance.
- • Protein synthesis is the most energy intensive process and requires the major proportion
of ATPs generated in the cell²². We observe a downregulation of ribosomes in the heart,
which is indicative of the lowered energy availability.
- • We also observed increased levels of the mitochondrial gatekeeper enzyme of glucose
oxidation (aerobic glycolysis and OXPPOS) pyruvate dehydrogenase kinase-1 and -4^{23,24}
(PDK-1, -4) in all tissues, with the highest level of upregulation being observed in the
heart (Supplemental figure). Increased PDK4 expression leads to the phosphorylation
and inactivation of pyruvate dehydrogenase (PDH), which transforms pyruvate, NAD⁺,
and coenzyme A into acetyl-CoA, carbon dioxide and NADH. A reduction in PDH activity
restricts the ability of mitochondria to utilize pyruvate, hence resulting in a lowered
oxygen consumption rate²⁵. This ensures reduced glucose utilization, and increased

glycolysis and fatty-acid oxidation. Despite its very short-lived protein half-life²⁶, we
observed a 6 log-2-fold change in the heart as early as 6 hours and was approximately 5
170 fold higher than other organs. This early upregulation of PDK4 is likely due to the higher
mitochondrial content of the heart. Interestingly, PDK4 is also upregulated in the hearts
of hibernating ground squirrels²⁷. Hence, PDK4 represents an evolutionary conserved
protein that regulates fuel usage during nutritional scarcity as seen in sepsis. Proteins
involved in the engagement of the foetal heart program (As described in comment 16
below), a metabolic characteristic of myocardial hibernation were upregulated. Although
we saw a minor upregulation of mitochondrial OXPHOS components in the heart, this is
indicative of the metabolic/mitochondrial hibernation and cardiac depression to tolerate
stress in sepsis¹⁵.

- • We have also described the changed markers for the activation of the untranslating
protein response (UPR) in the heart (see comment 2). The UPR is known for its
cytoprotective effects and can favour alternative pathways of ATP-generation and limit
OXPHOS dependent ATP-generation²⁸.
- • In the treatment cohort, we observed a reversion in the level of PDK4 in the heart and
elevation of pyruvate dehydrogenase (PDH) with late Gcc treatments at 8 hours. This
indicates an enhanced glucose utilization. This combined with reduced side effects on
the mitochondrial OXPHOS components (Fig. 6) could indicate the ability of mitochondria
to generate ATP in an OXPHOS-dependent manner. This, therefore, these represent a
potential metabolic recovery marker panel for testing mitochondrial resuscitation
therapies. Due to ethical constraints, we could not assess long term consequences of
treatments, and these represent a critical line of questioning. All in all, our findings in our
manuscript further emphasizes its importance as a resource for providing a base for
testing concepts pertaining to translational biology.

A few mitochondrial resuscitation strategies have been documented in a review by Protti et. al.¹⁷.
Nitric oxide (NO) generated via nitric oxide synthase (NOS) activity can inhibit ATP production.
The unspecific NOS inhibitor-NMMA (546C88) can resuscitate mitochondria by reducing NO
levels. In a phase III trial NMMA caused higher mortality, however, the drug was beneficial at
lower doses. Also, in a CLP mouse model, mitochondrial resuscitation using resveratrol was found
to improve cardiac mitochondrial function, reduce myocardial ultrastructural damage and
inflammation, without drastically improving survival²⁹. However, the study did not utilize any
antibiotic treated mice. Also, CLP manifests at the systemic level and affects all organs³⁰. Since
measurements for markers of organ damage, cytokine levels and bacterial load in organs other
than the heart were not provided, making it difficult to assess the organ-specific effects of
mitochondrial resuscitation. Both studies indicate that although restoration of mitochondrial
activity without understanding the details of the biology may be harmful. The strategy outlined in
our manuscript can aid in the understanding of the biology by -

- • Describe markers of multiple pathways that cause a downregulation of OXPHOS-
dependent ATP production.
- • Explain the global and organ-specific effects of mitochondrial resuscitation therapies.
- • The staging of metabolic dysfunction and the timing of mitochondrial resuscitation
therapies.

We have now added this to the discussion in line 459 as follows – “Cardiac dysfunction is an
important component in septic multiorgan failure and corresponding mitochondrial dysfunction
is linked to more severe outcomes. Our data show that hepatic mitochondrial components
including the OXPHOS are affected mainly in the liver but not in the heart in sepsis. In contrast,
antibiotics and glucocorticoids introduced cardiac mitochondrial fluctuations in septic animals
only. Our data shows that the late 8-hour combination therapy had minimal downregulation of

mitochondrial OXPHOS components and reduced the levels of the glucose oxidation gatekeeper
enzyme pyruvate dehydrogenase kinase 4 (PDK4) in the heart (Figure S6e). This indicates that the
mitochondrial function may be better preserved with late Gcc. Mitochondrial resuscitation could
benefit the treatment of septic shock, but due to the heterogeneity in organ and individual
responses, further studies must be performed to determine which patients would benefit the
most. The role of unintended mitochondrial dysfunction in post-sepsis sequelae, and
identification of surrogate marker that delineates cardiac mitochondrial status, also need to be
studied in relevant model systems and patients. Our strategy can be used to probe drug side
effects in other models.”

4. What is the mortality in the model? E. coli $10e^4$

Reply – Mortality cannot be assessed in our study due to current ethical constraints in Sweden.
The ethical guidelines used for the study explicitly state that the experiments should be
performed with a non-lethal dose of bacteria and weight loss of the mice should not exceed 15%.
The weight loss data have been displayed in supplemental figure S2B for the time course and
Figure 4B for the treatment cohort. In our pilot experiments we have determined that $10e4$
made the animals morbidly sick and resulting in loss of 15% bodyweight at 24 hours and $10e^5$
bacteria induced 100% mortality at 18 hours. Hence, we decided to use $10e^4$ in our
experiments.

5. What is UMAP2? Please specify what organ specific data was used to generate UMAP
projection of organs over time.

Reply – The full proteomics dataset from the time course experiment were used to generate the
UMAP. In addition, we have as a response to reviewer 2 added new supplement tables with the

data that was used to construct each Figure. The data presented in Fig. 1H UMAP was reduced to
2-dimensions for ease-of-visualization and UMAP2 is one of the 2 arbitrary dimensions of the
plot, akin to principal component analysis (PCA). UMAP stands for uniform manifold projection
and uses high-dimensional data to construct a low-dimensional representation that preserves
relationships present in the data³¹. UMAP has been used in defining cell types in mixed cell
populations and is useful in defining local and global relationships in large transcriptomics
datasets³². Therefore, to reduce the high dimensional proteomics data, and examine the local
and global relationships in organs and plasma over time, we chose to perform UMAP analysis.

6. Lung Histology – figure 2A – You may want to chose better representative photomicrographs
of the lung. The healthy lung looks like significant alveolar collapse with areas of PMN infiltration
and thickening of the alveolar capillary membrane – this “healthy’ lung is not a good example of
health. (I like the use of fibrin deposition to mark worsening lung injury – traditionally this is
associated with increased alveolitis – PMN infiltration – I don’t see that in the chosen images –
maybe you can pick a more representative photomicrograph that shows classic features of lung
injury)

Reply – We thank the reviewer for identifying the issue with image. We have now changed the
previous representative image for healthy lung for another that better depicts healthy/normal
morphology. The infiltration of neutrophil and deposition of fibrin at later stages of infection was
also observed in the mass spectrometry analysis. Due to problems in the sectioning, some of the
delicate structures like alveoli were damaged. We chose the previous control image because we
observed intact alveoli. However, we have now added new images that better illustrate
progressive lung damage over time.

7. This is not clear to me: The pattern and magnitude of DAPs subdivided organs into two groups.
that we refer to as responder organs (liver, leukocytes, and spleen) or bystander organs (lung,

heart, kidney, and plasma) – WHY? Sometimes “more” is just more – how do you know the
relative ‘weight/significance’ of specific DAP in a specific pathway (e.g. a rate limiting enzyme
would clearly have a much more profound effect on a path that multiple non-essential changes)
or on the trajectory of sepsis?

Reply – The reviewer raises a valid point. As presented in Figure 3B, we observed that certain
organs like liver, leukocytes, and spleen displayed a more balanced pattern of regulation in that
they contained both up and downregulated DAPs in response to sepsis and were referred to as
responder organs. The DAPs that were upregulated were immune effectors whereas the down
regulated DAPs contained out organ specific functions like fatty acid synthesis in liver. Whereas
other organs like lungs, heart, kidney, and plasma displayed mostly high number of upregulated
DAPs, like those belonging to the class of acute-phase plasma proteins and neutrophils were
referred to as bystander organs. As kindly pointed out in the comment by the reviewer we agree
that there is no evidence that organ responses can be labelled as responders or bystanders based
just on the pattern of regulation of DAPs and without the description of rate limiting enzymes.
Considering this, we have now withdrawn the terms responder and bystander organs. Instead,
we now refer to them as organs displaying bidirectional or unidirectional regulation of DAPs to
simply infer the pattern and magnitude of DAP regulation and not to refer functional status of the
organ directly.

We have added these changes in line 169 and now read as – “Based on the pattern and
magnitude of we observed organs displaying bidirectional (liver, leukocytes, and spleen) or
unidirectional (lung, heart, kidney, and plasma) regulation of DAPs”.

8. Page 11 - It is also unclear to this reviewer what the authors mean by: This subdivision may be
connected to the known activation of the leukocyte population (Fig 1E, CD11b+) and changes in
metabolisms during sepsis?

Reply – We apologise for the lack of clarity in the statement. We tried to justify the pattern of
DAP regulation (bidirectional regulation) in what was previously referred to as responder organs
(liver, leukocytes, and spleen) with known changes induced by neutrophil and bacterial influx.
Both liver and spleen are part of the reticuloendothelial system (RET) and are involved in the
capture and clearance of E. coli from the blood stream³³. Further that the neutrophils (CD11b+)
could actively traffic bacteria to the white pulp in the spleen³⁴ and also in turn induce a
downregulation of hepatic metabolism³⁵. However, in the absence of clear evidence as pointed
out in the previous comment, we have now withdrawn this statement from the manuscript.

9. Response to IL-1b and TLR4 – confined to organ tissues only – considering IL-1b and TLR4 share
the intracellular domain – it is not surprising that they also share response pattern – Maybe you
can select another receptor mediated pathway – that is confined to ‘organ’ specific response to
better illustrate your point?

We agree with this comment and now also highlight type-I interferon responses (type-I alpha,
and beta), regulation of the insulin pathway, and apoptosis as other processes that are confined
to the organs but not plasma. We have also described type-I interferon responses (type-I alpha,
and beta) as a pathway that is targeted by the glucocorticoids (Gcc) in figures 2C, 3G, 4E, 5A and
related supplemental figures in the first version of the manuscript.

We stated this below for the reviewer’s convenience in line 177 – “Other systemic proteome
changes confined to organ tissues included functions like altered Toll-like receptor-4 pattern
recognition receptor signalling pathway, insulin pathway regulation, and apoptosis (Fig. S3b).
Further, organs displaying unidirectional regulation of DAPs had minor downregulation in terms
of organotypic responses, as the above-mentioned changes represented a notable proportion of
the observed response (Fig. S3c).”

10. “breakpoint between 6-12 h where the increase in response is no longer linear” – What does

this time point signify? Is this the time it takes for 'de-novo' protein synthesis? Does this time
coincide with bacterial clearance?

Reply – Based on the comment above we realise that the term “breakpoint” can be
misinterpreted. Consequently, we have now removed this sentence and added text to convey
this in a better fashion (Line 166) – “Over the sepsis course, ramping up of responses (Fig. 3B),
and plateauing in the regulation of many of the DAPs at 12 hours (Figure S3a) was seen. DAPs
observed in blood plasma and the organs may be due to de novo synthesis, deposition, or
reduced clearance.” Our original line of thought was that this time point signifies a major ramping
up of host responses to counter the high bacterial load in organs. It also indicates the host
responses are possibly exhausted at the end of 12 hours as changes are not substantial for many
of the proteins at the end of 18 hours. The progressive increase in organ CFU also indicates
exhaustion of the immune responses. This is based on the following observations -

- • Based on the cluster analysis presented below (described in detail for our response to
comment14), where we show a high-level view of the order of events, innate immune
antibacterial measures like antimicrobial peptides and neutrophil proteins are
upregulated as early as 6 hours (new Supplementary figure, cluster 10). These are
presented as a heatmap and contains acute phase proteins like serum amyloids, hepcidin
and neutrophil granule proteins like cathelicidin and elastase. These proteins we observe
in organs are a result of de-novo synthesis, deposition, or reduced clearance. It is
especially hard to discern what is de novo synthesized and deposition in organs or
accumulation due to lack of clearance.
- • We would also like to clarify that in between 6 and 12 hours the host is overwhelmed by
the bacterial growth, which is up by a factor of $\log_{10}2$ CFUs (approximately) in all organs
(Figure 1) at 12 hours. The bacterial growth seems to up again by $\log_{10}2$ CFUs at 18 hours,

demonstrating the bacteria are increasing almost linearly. This indicates that host
measures are insufficient to counter bacterial growth in this model and we provide an
explanation as to why the bacteria is not cleared in our response to comment 18.

11. The subdivision of DAP types: Would it not be much clearer to subdivide DAPs into 2: Organ
specific versus shared between organs (include plasma as an organ). And for EACH subgroup have
2 categories – shared with plasma or not?

Reply – We would like to clarify that both views of the DAPs complement each other and provide
a more holistic view. Figure 3B captures the pattern of regulation of DAPs in terms of numbers at
the organ level. Our intention was to provide a simplistic representation of the magnitude of
DAPs in terms of ups and downs per organ compartment. The heatmaps in figures 3C and D
explore the more complex relationships of the data in terms of the shared and unique pattern of
DAP expression across organs (plasma included or not) and are data-driven. This is line with the
reviewer’s suggestion above. We have now rephrased the regulation of DAPs as being uni- or bi-
directional to just convey the pattern of regulation only.

12. “bystander organs were relatively refractory to intrinsic organotypic alterations” – what does
this mean? In the context of the number pf DAPs for each organ – I am not quite sure I
understand what you are trying to convey?

Reply –We have now changed the term bystander organ to organs displaying unidirectional
regulation of DAPs to depict the disproportionately high level of upregulations in contrast to the
lower number of downregulations. We have now changed this sentence in line 179 and it now
reads as follows – “Apart from this, organs displaying unidirectional regulation of DAPs had minor
downregulation in terms of organotypic responses, as the above-mentioned changes represented

a notable proportion of the observed response (Figure S3c).”

13. Can you provide an idea of the denominator – i.e. each organ must have a “portfolio” of
potentially ‘alterable proteins’ – for each organ the number of DAPs you found represents a
subsection of the available proteome (Figure 3B). What is that? Maybe rather than showing
absolute number of DaPs it would make more sense to present the data as the percent
proteome? And even then – would be important to know if the DAPs/available proteome –
specific categories of proteins were preferentially altered? To that end – Figure 1H makes a lot of
sense to me and is helpful in contextualizing the changes post-infection – but in the absence of a
denominator – it is unclear whether fold changes in basement membrane assembly related
proteins is because the basement membrane was truly targeted or is this because there is so
much basement membrane – or membrane associated proteins in the lung compared to other
organs. This is particularly important in the case of the lung for example – where over 40 cell
types are present – each expressing somewhere between 40-60% of their genome in relatively
small amounts because their numbers may be so small – compare this to something like heart or
liver – where the heterogeneity of the tissue is smaller and consequently the potential organ-
specific proteome smaller and all cells may be contributing to the same proteome

Reply - This is a very interesting idea that we did not consider in our first version of the
manuscript. The organ proteome can be envisioned to contain 4 different categories of proteins -
expressed-in-all, mixed, group-enriched, and tissue-enriched. We have added text to the
manuscript and included a new supplemental figure (Figure S3f) to highlight these findings.

Previous reports have shown that Most of the proteins belong to the expressed-in-all, and mixed
categories and are represented by essential proteins involved in metabolism and cell division.

While lesser number proteins belong to the group or tissue-enriched category and represent the
proteins specific to the specialised structure and function of the organs. During sepsis, we

observe an increase in the abundance of immune system mediators within all organs and a
rewiring of the organ-specific pathways. Therefore, the common denominator/alterable protein
portfolio consists of these 2 aspects in sepsis. We have now added this in the revised version of
the manuscript (Supplemental figure comment13_alterable_proteome) after attempting to
define the common denominator in the response in the following way:

- • Firstly, we identified tissue enriched genes by using the TissueEnrich package³⁶ with data
from healthy uninfected control animals as input. The teGeneRetrieval function was used
to define tissue-specific genes, using the algorithm from the Human Protein Atlas (HPA)
(Uhlén et al. 2015). We then broadly categorized the proteins into four groups:
expressed-in-all, mixed, group-enriched, and tissue-enriched as shown in the heatmap in
panel A.
- • In the next step, DAPs (both up and down) were then classified into previously mentioned
4 categories. Based on these categories of DAPS, we then proceeded to define the
following alterable denominator/protein-portfolio during sepsis (Figure S3f).
- • The stacked bar graphs in Figure S3f show the percentages of up and down DAPs in
organs. Many of the proteins remain unchanged overtime. At the end of 18 hours
maximum number of DAPs were seen in the organs. Most of the proteins that were
upregulated belonged to the expressed-in-all category (blue), with spleen and plasma
showing the highest level of enrichment. Whereas most of the downregulated proteins
belonged to the tissue- and group-enriched (red) with liver and leukocytes showing the
highest levels of downregulations (panel B).
- • Next, we employed UpSet plots to visualize the complexities of the regulation and
distribution of DAPs, and relationship between the organs. Each dot depicts DAP groups,
and the lines show shared DAPs across 2 or more organs. Hence, individual unconnected
dots in the plot depict uniquely upregulated DAPs in an organ. The intersect size at the

top denotes the number of DAPs per dot and the set size denotes the sharing of DAPs
between organs. The bar plot at the bottom of the data shows a breakdown of the DAPs
contained within each dot into the 4 categories of expressed-in-all (black bars), mixed
(red bars), group-enriched (green bars), and tissue-enriched (blue bars). The organs are
then ordered from smallest to the largest (top to bottom) in terms of numbers of
proteins and shared DAPs. Moving left across the plot shows uniquely regulated DAPs
(represented by individual dots) in organs and the opposite direction shows side of the
plot contained shared DAPs.

- • Most of the upregulated DAPs at 18 hours belonged to the expressed-in-all and mixed
category as displayed by the bar plots at the bottom of the plot, with the kidney
displaying the most of these and the lungs displaying the least. The proteins uniquely
upregulated in the kidneys mostly belonged to the mixed (>0.5%) and group-enriched
category (> 0.2%). Very few DAPs were in tissue enriched. Majority of the tissue-enriched
signal belonged to the shared DAPs between spleen and leukocytes and is likely because
of the highly similar composition (0.1%).
- • The downregulated DAPs were comprised of mostly group-enriched and expressed-in-all.
The leukocytes showed the highest number of downregulations, while the lungs again
contained the least. Most of DAPs were shared between 2 organs, with the plasma and
lungs containing the highest number of tissue-enriched DAPs. Therefore, it can be
concluded again that many of the responses seen in the downregulated DAPs were
group-enriched.
- • We had previously shown a manually curated heatmap showing some of the interesting
GO processes that we observe while sepsis. This was constructed by assigning GO terms
to the up and down regulated DAPs in our data. The upregulated GO terms were
associated were inflammation associated and were seen in all organs e.g., neutrophil

degranulation in all organs. Many of the downregulated processes were tissue specific
e.g., the downregulation of fatty acid metabolism in the liver. We have now added this in
the figure for the ease of contextualizing all our findings.

- • Based on this we can conclude that we find evidence for the following common
denominators–
a) even with substantial proteome changes the bulk of the proteome remains unaffected
in all organs.
b) The upregulated proteins belong to the more expressed in all and mixed across organs
and related to inflammatory response.
c) The downregulated DAPs are mostly driven by group-/tissue-enriched. These were
represented by liver and spleen linked to the rewiring of the metabolism and cell cycle
respectively.

We agree with the reviewer that the bulk proteome will not capture cellular
heterogeneity in various organs and will require single cell proteomics strategies. As of
now, this is laborious, requires highly specialised preparation methods and
instrumentation, and software pipelines to analyse the data are relatively new. This
represents an avenue for future studies. We have now added new text to highlight this as
a limitation and an avenue for new research in the discussion.

The suggested concept of a common denominator is an interesting one and we will use explore
this notion in more detail in our ongoing work.

We have added the following text to the manuscript as follows (Line 197)- “Alteration of DAPs in
terms of tissue-specific and common systems-wide denominators was also examined. DAPs were
classified into 4 categories – expressed-in-all, mixed, group-enriched, and tissue-enriched based
as described previously. Even though substantial changes to the proteome were observed, much

of the proteome remain unchanged (Figure S3f). Upregulated proteins mostly belonged to the
expressed-in-all and mixed categories and were involved in inflammatory responses. In contrast,
the downregulated DAPs belonged to the group-/tissue-enriched category. These could be linked
to the rewiring of the metabolism and cell cycle in the liver and spleen respectively.”

14. Figure 3D – rather than clustering by organ – would it not be much more clinically relevant to
cluster by function? Previous figures have already addressed the issue of change in number of
DAPs – I think this figure should underscore if different “functions” are activated over time – for
example – up-regulation of ligands, is followed by upregulation of intracellular signaling MAKs,
followed by Transcription factors, followed by cytokine/chemokines de novo gene synthesis ,
followed by upregulation of receptors and changes in cytoskeletal proteins (this is just an
example.....)

Reply – We thank the reviewer for their comment and have added the suggested changes to the
manuscript. As suggested by the reviewer, we have now in the revised version of the manuscript
reanalysed the data to enable the visualization of the general features associated with sepsis
progression. Cluster analysis of all the differentially expressed proteins independent of organ
origin was performed by assigning 10 clusters (Reviewer_1_Comment_14_Highleveldataview).
We then performed gene ontology (GO) analysis using metaspape (www.metaspape.org) on these
clusters. The data was sampled at 6, 12 and 18 hours post infection and is not sufficient to
capture all the events seen during sepsis. Nevertheless, we were able to define windows of
regulation that occur early at 6 hours or later at 12 and 18 hours. Our analysis revealed 4 broad
regulation groups and the following events commonly associated features of sepsis were
observed as listed below.

- • Among the early events with low fold changes upregulated at 6 hours (log₂fc between 1-
2.3), we observe increased abundance of cytokines like interleukin-1, type-I interferons,

neutrophil, MAPK activation and platelet proteins in clusters 1 and 4 as displayed in the
heatmap and protein-protein interaction networks (PPI networks). These terms highlight
early activation of innate immune mediators like platelets and neutrophils through
integrins and MAP kinase signalling and cytokine production. The data was also supported
by the metadata parameters such as cytokine levels neutrophils and platelet-neutrophil
complexes Reviewer_1_Comment_14_Highleveldataview).

• Early events with high fold changes upregulated at 6 hours (log2fc between 2.5-5) –
Proteins and GO terms in clusters 2, 7 and 10 were associated with early high fold
changes and included proteins/terms belonging to neutrophils, metalloprotease
activation, response to heat stress, interferon signalling and mitochondrial translation.
Elucidating on the importance of these, neutrophils are the most abundant cell type in
bacterial infections and therefore we observe several key neutrophil signatures early on.
This cluster also contains several interferon type-I components and is also associated with
sepsis. HSF-1 heat stress response and response to heat stress in cluster 7 are associated
with hyperthermia and indicate the early fever response. The upregulation in the
mitochondrial translation components may indicate compensatory mechanisms in
mitochondria due to stress in sepsis.

• Early downregulations with high fold changes at 6 hours were associated with cluster 5
(log2fc = -2.5). A reduction in the ribosomal and lipid metabolism components was seen,
which we have previously associated with the heart and liver respectively. These are
presented as protein-protein interactions (PPIs) are presented in figure
(Comment_14_highleveldataview). This showcases the metabolic remodelling events in
sepsis.

• Late events upregulated only at 12 hours (log2fc between 2.5-5) were contained in
clusters 3 and 6. Cluster 3 contained proteins associated with amino acid metabolism,

which may arise due to utilization of non-carbohydrate source like amino acids as sources
of energy. Cluster 8 contained type-I interferon- α , - β signalling, apoptosis, and pentose
phosphate pathway components. Examples of proteins from cluster 3 and 8 are
presented as PPIs in the figure. Related to the sequence of late events, we also display
leakage data in fig 2D-G to highlight appearance of metabolic proteins in plasma as a late
event at 12 and 18 hours. Most of the proteins in plasma are liver enriched and are in line
with the increase in liver damage associated marker alanine transaminase activity (ALT)
displayed in Figure S2. Cytoskeletal proteins are also seen to be up at 12 and 18 hours
and may be indicative general global cellular damage. These findings are also in line with
the increase in the marker of nonspecific cell damage, lactate dehydrogenase activity in
plasma (Figure S2).

In reference to Fig 3D, we realise the importance of cataloguing the global events and had
previously explored to view the data from a bird's eye perspective by condensing all regulated
proteins into a single data frame and then performing cluster analysis to reveal the broad
sequence of events independent of organ-specific profiles. However, this view excludes the organ
specific changes which are dictated by specialised function and subtle changes in the organs.
Therefore, we chose to display the figure in its current form, which is much more informative
about the organ-specific changes. Our main intention for figure-3 was to highlight the changes in
the number of DAPs and the pattern of regulation across organs to reveal tissue unique proteins
in all compartments, shared regulated proteins and uniquely regulated protein clusters. Fig3c
only reveals the tissue specific proteins and their presence in plasma. Substituting the current
figure panels in Fig3D for a high level non-condensed view will remove the overview of protein
regulations unique to organs and regulations shared between organs and plasma.

15. Please reference - hallmark proteins – how were these selected? How were they defined as
Hallmark (is this from metascape’s definition?)

Reply – We thank the reviewer for their comment. In this context we used the word hallmark as a
substitute for the infer proteins/pathways in liver known to be associated with sepsis such as IFN-
1 signalling, fatty acid metabolism and gluconeogenesis. We have highlighted the references for
these characteristic examples in the manuscript below –

- • The type I interferon signalling pathway is a target for glucocorticoid inhibition. Mol Cell
Biol 30, 4564-4574.
- • Reprogramming of basic metabolic pathways in microbial sepsis: therapeutic targets at
last? EMBO Mol Med 10.
- • Glucocorticoids in Sepsis: To Be or Not to Be. Front Immunol 11, 1318.

16. Figure 3K – not a lot of change in the expression of OXPHOS proteins – this is such a great
result! It really points to the fact that some proteins are not regulated at the abundance level. If
you pair this data with the increase in iNOS activation – or the expression of other mitochondrial
functional inhibitors it is quite clinically relevant. This could also be paired with data showing a
shift from aerobic to anaerobic glycolysis, and potentially the shift to expression of fetal proteins
in the heart – e.g. shift in expression away from aMHC and increase the expression of bMHC to
compensate for electron transport chain “poisoning” by NO (for example).

Reply –One of the key findings of the study is the upregulation of type-I beta interferons and TNF
signalling in organs, which can also cause mitochondrial dysfunction. We have also described the
shift to anaerobic glycolysis based on elevated levels of PDK4 (as stated in comment 4). Further,
to address the reviewer’s comments, we inspected the presence of iNOS and natural inhibitors of
mitochondrial function. We were unable to detect the presence of inducible nitric oxide

synthase (iNOS) in the heart. This may be due low levels of this proteins (below the limit of
detection) or that the protein is not amendable to mass spectrometry analysis. We did observe
reduced levels of Myh6 (Myosin heavy chain 6, alpha-MHC); however, we did not notice an
elevation of beta-MHC. The conversion from a- to b-MHC may be a result of long-term
remodelling and our short-term model is perhaps not sufficient to capture this change. Further,
we also found upregulation of proteins involved in glycogen synthesis, which is known to be
activated as a part of the stress activated foetal gene program³⁷. Proteins involved in the cellular
response to hypoxia and foetal major histocompatibility complex proteins were also elevated
(Reviewer_1_Comment_16_Fetal_cardiac_reprogramming).

17. Page 12: An example of how you contextualized changes of clinical relevant is the
LDLR/PCSK9/APOA1 and B shown in figure 3I – I think this makes the data immediately relevant
to translational biologists.

Reply – We thank the reviewer for highlighting the significance of the manuscript as a resource
for translational biologists.

18. Proteome “plateaus” at 12 hrs – begins to plateau at 8 hrs – what is the relationship between
this and bacterial clearance? From your data – in untreated mice there was very little bacterial
clearance that happened in organs in the first 18 hrs – did you do CFUs in blood? Can you
measure circulating endotoxin and its relationship to the proteome? (do you pick up endotoxin in
your proteomics analysis?)

Reply – We agree that there was very little bacterial clearance in the untreated animals. This
strain of E coli (O18:K1) is resistant to complement-mediated killing and allows the bacteria to
circumvent an important host defence mechanism³⁸, which may be a potential explanation for
this. We did not quantify CFUs in blood due to the high variability in our pilot studies.
Unfortunately, we also don't pick up endotoxin (LPS) due to its non-proteinaceous composition,

and our method setup is only suitable for detection of proteins. However, we used the CFU load
as a surrogate to perform correlation with the proteome (Reviewer_1_Comment_18_CFU-
Correlations). We performed correlation analysis on the full proteome dataset and log2CFU
(derived from CFU/g of tissue) using Spearman Rank correlation rho and saw 189 proteins that
correlated with increased bacterial load, out of which 89 correlated positively and 100 correlated
negatively. Enrichment analysis of the proteins revealed that acute phase response proteins that
are known to be enriched in plasma like SAA1 displayed high positive correlations. Epoxygenase
P450 pathway proteins involved in fatty acid metabolism enriched in the liver like Cyp2c67 was
negatively correlations. Taken together acute-phase plasma proteins and downregulated of fatty
acid metabolism in the liver showed strong positive and negative correlations with bacterial load.

19. Page 15 - "highlighting the importance of early treatment (Figure S4d and Figure S4e)" – I
would change this by stating the importance of early source control – since sepsis is the
dysregulated response to infection and meropenem does not treat the dysregulated response
but presumably eliminates the source.

Reply – We have now changed this to indicate early source control and added new text to the
manuscript.

The changed section now reads (Line 253) – "This highlights the importance of early source
control, and that the reversal is not absolute since meropenem does not treat the dysregulated
response directly but rather eliminates the source (Figure S4d and Figure S4e)."

20. Page 15 - In addition, our analysis revealed a third and more unexpected category – I would
eliminate this sentence – given all the side effects of antibiotics and steroids – it is not possible
that this comes as a surprise. Again – rather than underscoring the total number of proteins (this
may go in brackets) it would be more helpful to understand the proportion this represents of the
available proteome.

Reply – We have changed our statement in the main text of the manuscript accordingly. We
have also included the proportion of the proteins (15% approximately; 1055 out of 6895 total
proteins) that are identified as ‘side effect’ proteins for the treatments.

The changed abstract now reads –

“Mem introduced sepsis-independent perturbations in the mitochondrial proteome that were
counteracted by Gcc. Our findings establish the first-ever quantitative and organotypic
assessment of treatment effects of candidate therapies in relationship to dosing, timing, and
potential synergistic intervention combinations during sepsis.”

The changed main text reads (Line 261)-

“In addition, our analysis revealed a third category of proteins that were unaltered during sepsis
but significantly changed because of the interventions during septic conditions: sepsis-
independent effects (Fig. 4C-D). This is likely due to the side effects that are introduced by Mem
and Gcc in sepsis.”

21. Page 15 – Reversal of inflammatory response was most pronounced in the kidney – does this
reflect concentration of antibiotics and corticosteroids eliminated through the kidneys – do they
actually see more of the drug?

Reply- We cannot assess the pharmacokinetics of both drugs using MS-based proteomics so we
cannot comment on the concentrations of the interventions in the kidneys.

22. Page 18 - Functional enrichment shows that there is a substantial degree of inflammation,
such as neutrophil degranulation ($p = 10^{-37}$) and complement and coagulation cascades ($p = 10^{-$
29), that could not be reverted by any of the tested interventions. – This is nice data. And it
makes so much sense – given the mortality from sepsis remains around 20-40% despite our best

efforts – the 1092 proteins - what proportion of the sepsis proteome remains unchanged after
treatment? Are there categories of proteins that remain preferentially unchanged?

Reply – This is an interesting point brought up by the reviewer and 1092 represent approximately
16% (6895 total proteins) of the total proteome. In Figures 5A and B and Figure S5a of the first
version manuscript, we showcase the reversion of the pathways by treatments as 1-slope values.
As indicated in the scales for the plots, the degree of reversion of a given pathway by a treatment
in an organ is indicated by appearance of the colour green. Whereas non-reverted pathways are
indicated by the colour white. Specific categories of non-reverted pathways across all
compartments include acute phase response, platelet degranulation, and neutrophil
degranulation.

23. Page 18 - interventions can alter the levels of plasma biomarkers and organ damage markers
without affecting inflammation and organ damage. – I think here would be very relevant to link
back to the emerging concept of “treatable traits” – moving forward sepsis will be treated with a
‘mixture’ of agents that target multiple ‘treatable traits’

Reply - We agree with the reviewer for their comment and added text in the manuscript. Our
data indicate that the combination of glucocorticoids and antibiotics administered at 2 and 8
642 hours respectively can reduce the levels of biomarkers in plasma but without affecting the
643 inflammation or reversing liver damage. In contrast monotherapy with Gcc or Mem alone did not
reduce the leakage of tissue proteins. GccMem2h treatment mimics an immunocompromised
state due to the early administration of Gcc, followed by source control using antibiotics. We
were able to ascertain that this treatment had considerable detrimental effects based on both
physiological parameters and the combined proteome scoring in organs. This would have been
difficult to assess solely on the basis of plasma cytokine levels, flow cytometry on blood an organ

damage marker (Figure s4b). We have added new text as follows in line 416 to highlight how
treatments can help in defining treatable traits in sepsis – “In addition to sepsis endotypes, our
data demonstrate treatment-induced heterogeneity of organ responses, which should be
considered as a variable while defining such traits. Further, the strategy outlined here can help in
identifying and prioritizing such traits in preclinical models as multiple members of a pathway can
be validated.”

24. Page 19 - These DAPs were strongly linked to mitochondria and were enriched for electron
transport chain, mitochondrial transport, membrane organization, and translation – this is most
rewarding – exactly what would have been expected – nice!

Reply – We thank the reviewer for their positive feedback.

25. However, no clear signs of mitochondrial damage were observed except hydropic
mitochondria and perturbed cristae. (Takasu et al., 2013) – Other studies however have
documented by EM sepsis induced ultrastructural changes in mitochondria from mice that
developed moderate to severe myocardial depression post cecum ligation and puncture with
widespread swollen mitochondria with ruptured outer membranes, matrix paling and disrupted
cristae associated with vacuolation of the internal compartment DOI:

10.1097/CCM.0b013e31824e1370 – I suspect your H&E is not sensitive enough to identify
changes.

Reply – We have modified text in the manuscript. We would like to clarify that based on a
previously published report to indicate organ failure without signs of overt tissue damage. We
agree with the reviewer that other reports that report clear signs of mitochondrial damage in the
heart in sepsis and to this end we have now added the relevant reference and added textual

changes to the manuscript as follows in line 391– “Damage to mitochondria in the heart during
sepsis always results in the ultrastructural damage to cristae, even when damage is not overt.”

26. Discussion – please include in the discussion, a comment on findings re: mitochondria – this is
an old topic that is receiving emerging attention: (i) Bactericidal Antibiotics Induce Mitochondrial
Dysfunction and Oxidative Damage in Mammalian Cells (10.1126/scitranslmed.3006055); Do
antibiotics cause mitochondrial and immune cell dysfunction? A literature review
(10.1097/CCM.0b013e31824e1370); Antibiotics Friend and Foe: “From Wonder Drug to Causing
Mitochondrial Dysfunction, Disrupting Human Microbiome and Promoting Tumorigenesis”
(10.4236/ijcm.2018.93016);

Reply – We thank the reviewer for highlighting the resurgence in interest of studying
mitochondria. We have added this to the discussion as follows (line 460) – “Introduction of
mitochondrial dysfunction by antibiotics has been described previously but there has been a
recent resurgence in interest to revisit the topic.”

27. In the discussion – return to the point you make in the introduction – how does your study
define treatment-indicative biomarkers based on species conserved regulatory networks that can
be tracked in human studies and improve the translation of novel sepsis therapies from
preclinical models to humans.

Reply – We thank the reviewer for their comment and have added text to the discussion. Many of
the examples presented in the manuscript have been described are conserved across species and
have been described in humans. We believe that further testing using outbred mice, where
changes introduced by treatments over time are assessed while survival is accounted for must be
conducted to refine our findings and may help bridge the translational gap. In the discussion we
have added in line 444 - “Our data show that pathways describing lipid uptake via LDLR in the
liver, apolipoprotein B levels in plasma, mitochondrial dysfunction in the heart, interferon

responses in the kidney and platelet activation in the spleen can be envisioned as treatment-
indicative markers that are also relevant in humans. We believe that some of these can be
tracked in human studies to help bridge the translational gap.”

28. In the discussion – carefully attention should be paid to contextualizing why Gcc blunted
bacterial killing – and what could be potential “markers” that may be used to understand the
“proteostate” associated with high-risk of unwanted side effects.

Reply – We thank the reviewer for their comment. Gcc is a well described anti-inflammatory
agent and is known to downregulate a multitude of critical immune processes that govern
inflammation and bacterial clearance. We have highlighted many of these in the first version
manuscript –

- • Gc treatments resulted in reduced levels of pro-inflammatory cytokines like IL-6, TNF- α ,
and MCP-1 in plasma. These cytokines are critical markers for mounting the immune
response (Supplemental figure S4b).
- • Inflammatory processes like TLR and interferon signalling were downregulated across all
compartments.
- • Cytochrome b-245 heavy chain (Cybb), which is a component of the NADPH oxidase
(NOX2) was also reduced in leukocytes (Supplemental figure S4c). Cybb deficiency results
in defective ROS production and defective intracellular killing by leukocytes. Defective
Cybb is known to cause CGD (chronic granulomatous disease) which is characterised by
recurrent infections.

Some markers from the proteome data associated with risk of unwanted side effects that have
been described in the manuscript are as follows –

- • Cybb levels can convey the ability of leukocytes via ROS-based killing. Gcc2h and
Gcc2hMem8h displayed lowest levels of Cybb and were associated with high levels of
bacteria. This shows that host antibacterial mechanisms are needed to clear off the
infection in addition to antibiotics (Figure 3 and Figure S4b).
- • Glucocorticoids can influence lipid metabolism. The lipid metabolism in plasma and liver
as governed by the PCSK9, LDLR and Apoe as described in Figure 5E can also be used.
- • We have also dedicated figure 6 to describe the side effects visible in the heart and this
can also be used to describe the side effects.

29. In the discussion – special attention should also be paid to the inability of steroids and/or
antibiotics to “revert” the septic-phenotype – your data suggests this is not a feature of host
heterogeneity (quality of the variability), but rather a feature of complexity (quantity – number of
variables). If your data is correct – even correctly assigning patients into specific endotypes will
not allow us to completely reverse the septic phenotype because current pharmacotherapy
cannot address the entirety of the complexity. Accordingly new treatments are urgently needed -
that adequately tackle major gaps – treatable traits – that are not being treated with our current
approaches.

Reply –We agree with the reviewer that the level of reversion is minor by even the best possible
therapy in our study represented by MemGcc8h. This is related to the complexity of the
responses in individual mice, as the mice included in our studies are genetically homogenous. The
heterogeneity in sepsis has been described to be multidimensional. However, the heterogeneity
in terms of organ responses to treatments is unaccounted. We believe that our data indicate that
treatments have a preferential rescue effect in sepsis. For example, that the spleen benefitted
the most from the treatment in terms of reversions and proximity to the naïve animals. Other
organs such as heart and plasma remain extremely variable. Further studies are needed to

address to characterise treatment induced heterogeneity in organs and consider these as a
factor.

We did not study the long-term effects of the reshuffling of the proteome in terms of recovery
and mortality. Further long-term studies are needed to assess the key events such as time taken
to completely revert the proteome to resemble naïve mice and assess mortality.

We have added text to the discussion as follows (line 413) –

“Recently, Sepsis was categorised into endotypes using plasma biomarkers and clinical outcomes
to better understand the heterogeneity of treatment effects. Based on these findings, the
concept of treatable traits in sepsis is the next step that defines actionable therapeutic targets. In
addition to sepsis endotypes, our data demonstrate the treatment-induced heterogeneity of
organ responses and should be considered as a variable while defining such traits. Further, the
strategy outlined here can help in identifying and prioritizing such traits in preclinical models as
multiple members of a pathway can be validated.”

30. In the discussion – It would be important to also address the issue of iatrogenicity and hard
and how your data may be used in the development of harm reduction strategies.

Reply – We have added text in the discussion in response to comment 4. The iatrogenic effects
on the heart was the strongest phenotypes we picked up using the scoring system. We have in
our previous responses also commented on why this occurs and that timing of resuscitating
mitochondrial drugs are critical. These need to be explored meticulously in future studies. The
strategy described in the manuscript can be performed with either proteomics or transcriptomics
can capture side effects more efficiently should then be applied.

Reviewer #2 (Remarks to the Author):

Combining conventional markers, histology, proteomics, and systems biology, the authors

performed pharmacoproteomic analysis on a murine E. Coli sepsis model by scoring time-
dependent treatment impact after Mem and Gcc were administrated. They found three organ-
specific proteome response patterns, several novel elements of the host response during the
course of sepsis and key therapeutic interventions. Overall, this manuscript is well written and
comprehensive. Below are some minor suggestions, mainly about the proteomic data, for the
authors to consider before publication.

1. Please complete the code for “under accession code XXXXX”.

Reply- We have now added the accession codes associated to the dataset (PXD036832,
PXD036847) to the manuscript.

2. Since this paper is based on the proteomic data, I would suggest providing a bit detailed
description about data QC such as missing value rate (e.g., per sample and overall), batch/study
design, replicate information if any, etc.

Reply – We thank the reviewer for their comment. Accordingly, we have added a new
supplemental figure (Figure S1b) that provides a detailed description of quality control (QC)
parameters such as missing values, replicate information, and batch/study design.

3. “Missing values were imputed with the assumption of left-censored data missing not at
random.” The authors could explain here a bit more why this assumption holds for their
proteomic data. In fact, proteomic data are subject to both missing not at random and missing
(completely) at random. And it is known that SWATH-MS data have a lot of missing values.

Reply - Missing values in our DIA data is strongly associated with low abundant proteins
(Supplemental figure S1b). Our assumption is that missingness is due to precursors/proteins
being below the MS detection limit in that sample and hence missing not at random. This is in line
with recent publications (ref 1; all references added at the bottom of the comment) and software

documentation for DIA data extraction software (ref 2,3). We acknowledge that imputation of
missing values in MS-based proteomics is a complex issue and that a variety of published
algorithms and strategies exist (ref 1, 4, 5). But to our knowledge there is no unbiased way of
determining the type of missingness nor is there any consensus on the best imputation method.
Additionally, our DIA data was post-processed with TRIC (ref 6), this alignment tool has been
shown to substantially decrease missingness of DIA data.

References

1<https://journals.plos.org/plosone/article?id=10.1371/journal.pone.0249771> Ref

2<https://github.com/vdemichev/DiaNN#frequently-asked-questions> Ref

3https://biognosys.com/content/uploads/2022/12/Spectronaut17_UserManual.pdf Ref

4<https://www.nature.com/articles/s41598-022-04938-0> Ref

5<https://academic.oup.com/nar/article/48/14/e83/5856122> Ref

6<https://www.nature.com/articles/nmeth.3954>

4. The authors mentioned that batch-effect correction approach was to utilize protein log₂
foldchange in relation to naïve samples per experiment. But there is no information about how
batch correction was done and how the experiment/batch was designed.

Reply- We apologise for adding this statement to the methods as we have not performed batch
correction across experiments. We have treated each experiment separately. Based on this we
have now removed the statement from the methods section.

5. According to Fig.1D, different organs have different numbers of proteins. Does this mean that
Fig. 1H was plotted with the overlapping/common proteins? The same question for all other plots
where multiple organs are present.

Reply – We Thank the reviewer for their comment. Accordingly, we have modified the methods
section and would like to clarify that fig1H contains all identified proteins and if a protein was not
detected in an organ (missing in an organ) it was assigned as zero. A similar strategy was adopted
for other figures where multiple organs are present.

6. Probably an overview heatmap showing the proteomic data of all organs will reveal organ-
specific missingness pattern, organ-specific proteins, house-keeping proteins, and proportion of
missingness.

Reply – We have added a new supplemental figure (Figure S1b) that shows quality control and
data pre-processing of the proteomic data and includes protein intensity heatmaps in panel B.
Aggregated missingness is also shown in a separate panel F.

Reviewer #3 (Remarks to the Author):

This manuscript is ambitious and will have a wide breath impact on the longitudinal
characterization of murine models of sepsis. The text is well-written and despite the highly
technical scope of approach and results, it will be easy to understand by non-experts in MS-
proteomics.

Concerns:

1. The authors wrote: "Our findings establish the first-ever quantitative and organotypic
assessment of treatment effects of candidate therapies in relationship to dosing, timing, and
potential synergistic intervention combinations during sepsis"

This claim is more pompous than accurate. The effort presented here is unique in its kind given

the amount of proteomics data collection, but is not the first longitudinal study of proteome
remodeling in septic mice. It is also not the first study to use proteomics to assess organ damage
or response drugs in murine sepsis. The authors should tone down their claims and acknowledge
the following papers:

PMID: 31568522; PMID: 32037835; PMID: 35349828; PMID: 35913192; PMID: 31604940.

Reply- We recognise that previous studies detail the longitudinal effects of sepsis, including
damage markers in plasma, changes in organs with infection alone or exploration of the plasma
proteome with antibiotics only We have now also added all suggested references to the
introduction. To the best of our knowledge none of the studies, including the ones above, have
so far attempted to characterise the effects of treatment on the organs, especially combination
therapies, dosing, timing and provide a scoring system that captures iatrogenicity of drug
candidates. Accordingly, we have now modified the abstract as follows – “We provide a novel
base for the quantitative and organotypic assessment of treatment effects of candidate therapies
in relationship to dosing, timing, and potential synergistic intervention combinations during
sepsis.”

2. The area of computational analysis of quantitative proteomics is currently well-established and
most experts in the field use the same software, which has been stress-tested and shown to
deliver accurate results. The authors of this manuscript should explain why they chose to use
non-stream computational tools.

Reply – We would like to clarify that the data were generated using previously well-established
software and workflows in the first version of the manuscript. For the convenience of the
reviewer, we have stated this briefly below as follows-

- • MS data were searched using FragPipe (v12.2), MSFragger (v2.4) and Philosopher (v3.2.3). The spectral library from was compiled from the FragPipe output using Spectrast

(v5.0), and msproteomicstools (v0.11.0). DIA data files (n=568) were analyzed against the
RT and main spectral libraries with OpenSwathWorkflow (v2.4). The OpenSwath data was
scored with PyProphet (v2.1.3) utilizing 1% FDR on protein and peptide levels. Finally, the
peakgroups were aligned with TRIC from msproteomicstools (v0.11.0).

- • Precursor intensities were summed into protein intensities and the data was normalized
with function justvs() from package vsn (v3.52.0) per sample type and mouse
experiment. Proteins with missing values were imputed with base R stats::runif() function
with the assumption of left-censored data missing not at random. Differential abundance
testing was performed with R package limma (1.11.1) or with base R function
stats::t.test(). Cut-offs were foldchange > ±1.5 and adjusted (with method of Benjamini,
Hochberg) p-value < 0.05.
- • Functional and pathway enrichment analysis was performed with Metascape (v33) using
the web interface (<https://metascape.org/>) and the 'Express Analysis of Multiple Gene
Lists' workflow.
- • UMAP projections were generated with R package uwot (v0.1.4). Heatmaps were
generated with R package ComplexHeatmap (v2.0.0) using ward.D2 cluster analysis. The
MitoCarta3 mouse database was used for defining mitochondrial proteins and subdivision
into functional groups.
- • Data analysis was performed, and graphical outputs were generated using R (3.6) using
the R package Tidyverse (1.3.0) and Bioconductor package manager BiocManager (v3.9).
- • Networks included in the manuscripts were generated as part of the standard output of
metascape. Intervention networks were generated using cytoscape.
- • The MitoCarta3 mouse database was used for defining mitochondrial proteins and
subdivision into functional groups.

• Intervention score plots were generated using ggplot2 (part of tidyverse1.3.0). Cut-offs
were foldchange > ±1.5 and adjusted (with method of Benjamini, Hochberg) p-value <
0.05 for defining intervention effects.

• To improve the transparency of our R analyses we have uploaded R notebooks (.html and
.rmd versions) to a Zenodo repository accessible via
<https://doi.org/10.5281/zenodo.7755876>. The notebooks show calculations on MS-data
for both time course and intervention experiments. Source data will be added to the
repository in case of manuscript acceptance. For example, custom scripts were used for
data extraction from the metascape outputs.

All details are provided in the methods and material section.

3. The manuscript suffers from a lack of orthogonal validation of at least some of the quantitative
calculations. This is important due to the use of uncommon computational tools.

Reply – We thank the reviewer and would again like to reiterate that we have used previously
well-established software to generate all our quantitative calculations as stated in the comment
above. In the first version of the manuscript, we have also validated the intervention scoring in a
second independent cohort provided in supplemental figure S4e.

4. There are typos here and there that should be taken care of.

Reply- We apologise for this and have now proofread the manuscript to remove the typos.

Liu, X. & Green, R. M. Endoplasmic reticulum stress and liver diseases. *Liver Res* **3**, 55-64,
doi:10.1016/j.livres.2019.01.002 (2019).

Baird, T. D. & Wek, R. C. Eukaryotic Initiation Factor 2 Phosphorylation and Translational
Control in Metabolism. *Adv Nutr* **3**, 307-321, doi:10.3945/an.112.002113 (2012).

Jiang, D. *et al.* ATF4 Mediates Mitochondrial Unfolded Protein Response in Alveolar
Epithelial Cells. *Am J Respir Cell Mol Biol* **63**, 478-489, doi:10.1165/rcmb.2020-01070C
(2020).

Pakos-Zebrucka, K. *et al.* The integrated stress response. *Embo Rep* **17**, 1374-1395,
doi:10.15252/embr.201642195 (2016).

Grootjans, J., Kaser, A., Kaufman, R. J. & Blumberg, R. S. The unfolded protein response in
immunity and inflammation. *Nat Rev Immunol* **16**, 469-484, doi:10.1038/nri.2016.62 (2016).

Steffen, K. K. & Dillin, A. A Ribosomal Perspective on Proteostasis and Aging. *Cell Metab* **23**,
1004-1012, doi:10.1016/j.cmet.2016.05.013 (2016).

Somogyvari, M., Khatatneh, S. & Soti, C. Hsp90: From Cellular to Organismal Proteostasis.
*Cells* **11**, doi:10.3390/cells11162479 (2022).

Hirano, Y. *et al.* A heterodimeric complex that promotes the assembly of mammalian 20S
proteasomes. *Nature* **437**, 1381-1385, doi:10.1038/nature04106 (2005).

Le Tallec, B. *et al.* 20S proteasome assembly is orchestrated by two of chaperones in yeast
distinct pairs and in mammals. *Mol Cell* **27**, 660-674, doi:10.1016/j.molcel.2007.06.025
(2007).

Liu, M. & Dudley, S. C., Jr. Role for the Unfolded Protein Response in Heart Disease and
Cardiac Arrhythmias. *Int J Mol Sci* **17**, doi:10.3390/ijms17010052 (2015).

Eisele, Y. S. Amyloidosis by Bacterial Infection in Critically Ill Patients? *Am J Respir Crit Care*
*Med* **198**, 1475-1476, doi:10.1164/rccm.201809-1777ED (2018).

Mesgarzadeh, J. S., Buxbaum, J. N. & Wiseman, R. L. Stress-responsive regulation of
extracellular proteostasis. *J Cell Biol* **221**, doi:10.1083/jcb.202112104 (2022).

Ali, L. *et al.* Common gene variants in ASGR1 gene locus associate with reduced
cardiovascular risk in absence of pleiotropic effects. *Atherosclerosis* **306**, 15-21,
doi:10.1016/j.atherosclerosis.2020.07.001 (2020).

Islam, M. M., Hlushchenko, I. & Pfisterer, S. G. Low-Density Lipoprotein Internalization,
Degradation and Receptor Recycling Along Membrane Contact Sites. *Front Cell Dev Biol* **10**,
826379, doi:10.3389/fcell.2022.826379 (2022).

Singer, M. Mitochondrial function in sepsis: acute phase versus multiple organ failure. *Crit*
*Care Med* **35**, S441-448, doi:10.1097/01.CCM.0000278049.48333.78 (2007).

Pravda, J. Metabolic theory of septic shock. *World J Crit Care Med* **3**, 45-54,
doi:10.5492/wjccm.v3.i2.45 (2014).

Protti, A. & Singer, M. Bench-to-bedside review: potential strategies to protect or reverse
mitochondrial dysfunction in sepsis-induced organ failure. *Crit Care* **10**, 228,
doi:10.1186/cc5014 (2006).

Singer, M., De Santis, V., Vitale, D. & Jeffcoate, W. Multiorgan failure is an adaptive,
endocrine-mediated, metabolic response to overwhelming systemic inflammation. *Lancet*
**364**, 545-548, doi:10.1016/S0140-6736(04)16815-3 (2004).

Ashare, A. *et al.* Insulin-like growth factor-1 improves survival in sepsis via enhanced hepatic
bacterial clearance. *Am J Respir Crit Care Med* **178**, 149-157, doi:10.1164/rccm.200709-
1400OC (2008).

Moller, N. & Jorgensen, J. O. Effects of growth hormone on glucose, lipid, and protein
metabolism in human subjects. *Endocr Rev* **30**, 152-177, doi:10.1210/er.2008-0027 (2009).

Faraj, M. LDL, LDL receptors, and PCSK9 as modulators of the risk for type 2 diabetes: a focus
on white adipose tissue. *J Biomed Res* **34**, 251-259, doi:10.7555/Jbr.34.20190124 (2020).

Buttgerit, F. & Brand, M. D. A Hierarchy of Atp-Consuming Processes in Mammalian-Cells.
*Biochem J* **312**, 163-167, doi:DOI 10.1042/bj3120163 (1995).

Kaplon, J. *et al.* A key role for mitochondrial gatekeeper pyruvate dehydrogenase in
oncogene-induced senescence. *Nature* **498**, 109-+, doi:10.1038/nature12154 (2013).

Gray, L. R., Tompkins, S. C. & Taylor, E. B. Regulation of pyruvate metabolism and human
disease. *Cell Mol Life Sci* **71**, 2577-2604, doi:10.1007/s00018-013-1539-2 (2014).

- Shimada, B. K. *et al.* Pyruvate-Driven Oxidative Phosphorylation is Downregulated in Sepsis-
Induced Cardiomyopathy: A Study of Mitochondrial Proteome. *Shock* **57**, 553-564,
doi:10.1097/Shk.0000000000001858 (2022).
- Crewe, C., Schafer, C., Lee, I., Kinter, M. & Szweda, L. I. Regulation of Pyruvate
Dehydrogenase Kinase 4 in the Heart through Degradation by the Lon Protease in Response
to Mitochondrial Substrate Availability. *Journal of Biological Chemistry* **292**, 305-312,
doi:10.1074/jbc.M116.754127 (2017).
- Buck, M. J., Squire, T. L. & Andrews, M. T. Coordinate expression of the PDK4 gene: a means
of regulating fuel selection in a hibernating mammal. *Physiol Genomics* **8**, 5-13,
doi:10.1152/physiolgenomics.00076.2001 (2002).
- Zhu, L., Luo, X., Fu, N. & Chen, L. Mitochondrial unfolded protein response: A novel pathway
in metabolism and immunity. *Pharmacol Res* **168**, 105603, doi:10.1016/j.phrs.2021.105603
(2021).
- Smeding, L. *et al.* Salutary effect of resveratrol on sepsis-induced myocardial depression. *Crit*
*Care Med* **40**, 1896-1907, doi:10.1097/CCM.0b013e31824e1370 (2012).
- Rumienczyk, I. *et al.* Multi-Organ Transcriptome Dynamics in a Mouse Model of Cecal
Ligation and Puncture-Induced Polymicrobial Sepsis. *J Inflamm Res* **14**, 2377-2388,
doi:10.2147/JIR.S307305 (2021).
- McInnes, L., Healy, J. & Melville, J. UMAP: Uniform manifold approximation and projection
for dimension reduction. . *Stat. Mach. Learn. arXiv preprint arXiv:1802.03426* (2018).
- Becht, E. *et al.* Dimensionality reduction for visualizing single-cell data using UMAP. *Nat*
*Biotechnol*, doi:10.1038/nbt.4314 (2018).
- Pelkonen, S. & Pluschke, G. Roles of spleen and liver in the clearance of Escherichia coli K1
bacteraemia in infant rats. *Microb Pathog* **6**, 93-102, doi:10.1016/0882-4010(89)90012-0
(1989).
- Juzenaite, G. *et al.* Lung Marginated and Splenic Murine Resident Neutrophils Constitute
Pioneers in Tissue-Defense During Systemic E. coli Challenge. *Front Immunol* **12**, 597595,
doi:10.3389/fimmu.2021.597595 (2021).
- Crespo, M. *et al.* Neutrophil infiltration regulates clock-gene expression to organize daily
hepatic metabolism. *Elife* **9**, doi:10.7554/eLife.59258 (2020).
- Jain, A. & Tuteja, G. TissueEnrich: Tissue-specific gene enrichment analysis. *Bioinformatics*
**35**, 1966-1967, doi:10.1093/bioinformatics/bty890 (2019).
- Taegtmeier, H., Sen, S. & Vela, D. Return to the fetal gene program: a suggested metabolic
link to gene expression in the heart. *Ann N Y Acad Sci* **1188**, 191-198, doi:10.1111/j.1749-
6632.2009.05100.x (2010).
- Wooster, D. G., Maruvada, R., Blom, A. M. & Prasadarao, N. V. Logarithmic phase Escherichia
coli K1 efficiently avoids serum killing by promoting C4bp-mediated C3b and C4b
degradation. *Immunology* **117**, 482-493, doi:10.1111/j.1365-2567.2006.02323.x (2006).

Reviewer_1_Comment_2 (Added in manuscript as Fig. S3e): Mechanisms of proteostasis in sepsis.

We observed a major reduction of cytosolic ribosomal components in the heart that constitute the first step of proteostasis. Our analysis also revealed a loss of proteostasis with progression of sepsis. Molecular sensors for detection of unfolded proteins in the endoplasmic reticulum like Inositol-requiring protein 1 (IRE1a) and interferon-induced double-stranded RNA-activated protein kinase (EIF2AK2), were up in all organs. Also, different isoforms of HSP90 (HSP90aa1, HSP90b1 and HSP90ab1) that are responsible for protein folding, and proteasome assembly chaperones (PSMG1-4) that promote the assembly of the 20S proteasome were elevated. In order to protect the secreted proteome, several extracellular chaperones like haptoglobin (HP), alpha-2-macroglobulin (A2M), clusterin (CLUS), heat shock protein beta-1 (HSPB1), progranulin (GRN), and serum amyloid P-component (APCS) in plasma and tissues were induced and could represent an active countermeasure for the formation of amyloid fibrils. The levels of ASGR1 responsible for clearance of asialylated proteins were downregulated in the liver during sepsis. In summary, the heart was strongly enriched for markers of perturbed proteostasis compared to others. Panel A) shows these markers in the time course experiments and B) shows the levels in intervention experiments.

Time Course

Intervention studies

Reviewer_1_Comment_4 (Added in manuscript as Fig. S6e): Upregulation of pyruvate dehydrogenase kinases (PDKs)

Heatmaps showing the regulation of PDKs. Here we show the levels of the mitochondrial matrix gate keeper enzyme pyruvate dehydrogenase kinase-4 and other isoforms (2-4) in the time course and intervention experiments. Increased PDK4 expression leads to the phosphorylation and inactivation of pyruvate dehydrogenase (PDH) that regulates the entry of pyruvate into the Krebs cycle and oxidative phosphorylation (OXPHOS). A reduction in PDH activity restricts the ability of mitochondria to utilize pyruvate, hence resulting in a lowered oxygen consumption rate. This ensures increased glycolysis and fatty-acid oxidation (Ref. 1-3). Despite its very short-lived protein half-life (Ref.4), we observed a 6 log-2-fold change in the heart as early as 6 hours and was approximately 5-fold higher than other organs. This early upregulation of PDK4 is likely due to the higher mitochondrial content of the heart. In the treatment cohort, we observed a reversion in the level of PDK4 in the heart and elevation of pyruvate dehydrogenase (PDH) with late Gcc treatments at 8 hours. This combined with the reduced side effects in the mitochondrial OXPHOS components described in figure 6 indicates improved utilization of glucose by mitochondria and a reduction of mitochondrial dysfunction.

- 1 Kaplon, J. et al. A key role for mitochondrial gatekeeper pyruvate dehydrogenase in oncogene-induced senescence. *Nature* 498, 109-+, doi:10.1038/nature12154 (2013).
- 2 Gray, L. R., Tompkins, S. C. & Taylor, E. B. Regulation of pyruvate metabolism and human disease. *Cell Mol Life Sci* 71, 2577-2604, doi:10.1007/s00018-013-1539-2 (2014).
- 3 Shimada, B. K. et al. Pyruvate-Driven Oxidative Phosphorylation is Downregulated in Sepsis-Induced Cardiomyopathy: A Study of Mitochondrial Proteome. *Shock* 57, 553-564, doi:10.1097/Shk.0000000000001858 (2022).
- 4 Crewe, C., Schafer, C., Lee, I., Kinter, M. & Szveda, L. I. Regulation of Pyruvate Dehydrogenase Kinase 4 in the Heart through Degradation by the Lon Protease in Response to Mitochondrial Substrate Availability. *Journal of Biological Chemistry* 292, 305-312, doi:10.1074/jbc.M116.754127 (2017).

A

B

C

Reviewer_1_Comment_10 Ramping up and plateauing of responses of DAPs at 12 hours

A) Global Cluster analysis of the proteome. All significant proteins were subjected to cluster analysis without subdivision into organs. Clusters-2, 4, 7 and 10 showed higher early fold changes. B) Protein-protein interaction (PPI) network for some of the proteins in cluster 10 as generated by metascape.

C) Heatmap showing proteins from cluster10.

Reviewer_1_Comment_14
Cluster analysis of the global proteome response.
 A) All significantly regulated proteins (p < 0.05) were clustered into 10 clusters. The red line in each cluster represents the mean Log2Fc. Metascape GO analysis was then performed on each cluster and the heat map of the GO terms are shown in B) and the TRUST transcription factor analysis are shown in C). Protein-protein interaction networks are shown for clusters in D).

A) Major histocompatibility complex

B) Alpha-myosin heavy chain

C) Glycogen synthesis

D) Cellular response to hypoxia

Reviewer_1 Comment_16 Foetal reprogramming profile of the septic heart. A) Panels showing proteins involved in major histocompatibility complex. Proteins involved in the cellular response to hypoxia and foetal major histocompatibility complex proteins (H2-K1, H2-Q4) were elevated. The adult MHC complex component H2-Q10 was reduced in the heart.

B) Alpha-myosin heavy chain levels were reduced over time and has been described to be a part of the foetal reprogramming stress response.

C) Components of glycogen synthesis are upregulated over time. Here we show an upregulation in the levels of Gsk3a, Ppp2r1a, Pgm2l1, Ugp2, Pygl and Gys1.

D) Several markers belonging to the GO term cellular response of hypoxia were also upregulated and included several proteasome components (Psmc subunits, Psm subunits and Psmn subunits). We also saw the upregulation of the hypoxia-inducible factor prolyl hydroxylase 2 (HIF-PH2) or Egn1. These indicate the underlying state of hypoxia within the septic heart.

A

B

CFU Time

C

CFU Positive correlation enrichments

CFU Negative correlation enrichments

Reviewer_1_Comment_18 Proteome-CFU correlations over time.

A) Bar plots showing the distribution of Spearman's rank correlation (rho) of the proteome data with time. 86 proteins co-related with increased bacterial load, out of which 46 correlated positively and 40 correlated negatively.

B) Examples of positive and negative correlations over time with elevated bacterial load (Colony Forming Units; CFU). Acute phase response proteins that are known to be enriched in plasma like SAA1 displayed high positive correlations, whereas proteins involved in fatty acid metabolism enriched in the liver like Cyp2c67 had high negative correlations. C) GO analysis using metasplice of positive- and negative-correlations with CFU. CFU positive correlations enrichments indicate an elevation of inflammatory components like acute phase reactants and cytokines. CFU negative correlations included several liver-enriched fatty acid metabolic processes.

A

180805							191210						200319			
	Heart	Kidney	Liver	Lungs	Plasma	Spleen	Heart	Kidney	Leukocytes	Liver	Plasma	Spleen	Kidney	Leukocytes	Plasma	Spleen
Naive	5	5	5	5	4	5	6	6	5	6	6	6	6	5	6	6
Inf_6h	6	6	6	6	6	6	6	6	5	6	6	6	3	3	3	3
Inf_12h	6	6	6	6	6	5	6	6	5	6	5	6	5	5	5	5
Inf_18h	6	6	6	6	6	6	6	7	7	7	7	6	5	5	5	5
Inf_Gcc2h	6	6	6	6	6	6	6	6	5	6	5	6	5	5	5	5
Inf_Gcc2h+Mem8h	7	7	7	7	7	7	7	7	6	7	6	7	5	5	5	5
Inf_Gcc8h+Mem8h	6	6	6	6	6	6	6	6	5	6	6	6	5	5	5	5

B

C

D

E

F

G

Reviewer_2_comment_2 (Added in manuscript as Fig. S1b) Quality control and preprocessing of DIA data. The DIA data is divided into three datasets – 180805 sepsis time course, 191210 & 200319 interventions. A) The sample groups and replicate count of each experiment and group. B) Heatmaps of protein intensity across experiments. C) Protein ID count per sample of unfiltered data D) Protein ID count per sample after applying missing value filter (the missing value filter requires at most 1 missing value in 1 sample group). E) Distribution of coefficient of variation (CV) of protein intensities per organ. Color indicates CV of raw intensities and CV of intensities after applying variance stabilizing normalization (VSN). F) Median proteins intensities per experiment & organ are discretised into 8 categories, y-axis Intensity.Bin, and the fraction of missing values per protein and organ boxplots are on the x-axis Fraction.Imputed. G) Intensity distribution of measured and imputed protein intensities per experiment.